# Saturated fatty acids induce lipotoxicity in lymphatic endothelial cells contributing to secondary lymphedema development

Karina P Gomes 🔟[1], Jacob Korodimas 🔟[1], Emily Liu 🔟[1], Nirav Patel[1], Xiaoyan Yang 🔟[1], Susan Goruk 🔟[2], Jaqueline Munhoz 🔟[2], Catherine J Field 🔟[2] & Spencer B Gibson 🔟[1✉]

## Abstract

**Lymphedema is a chronic lymphatic disorder characterized by persistent tissue swelling, pain, and recurrent infections, often secondary to cancer treatment, surgery, or obesity. Obesity-associated increases in saturated fatty acids (SFAs) have been linked to lipotoxicity. In this study, patients with secondary lymphedema showed a significantly lower plasma polyunsaturated to saturated fatty acid (PUFA/SFA) ratio compared to BMI-matched controls. Stearic acid, a common dietary SFA, induced apoptosis, oxidative stress, and endoplasmic reticulum (ER) stress in human lymphatic endothelial cells. In a mouse model, a short-term high-SFA diet was used to lower the plasma PUFA/SFA ratio, which worsened tail swelling, oxidative stress, ER stress, and tissue damage following lymphatic injury. Switching to a standard chow diet after surgery prevented these effects. Patients with lymphedema also exhibited elevated levels of fatty acid-binding protein 4 (FABP4), a lipid chaperone associated with metabolic stress. FABP4 inhibition reduced stearic acid-induced cell death in vitro and mitigated tissue damage in vivo. These findings suggest a pathogenic role for SFAs and support dietary modulation and FABP4 inhibition as potential therapeutic strategies for lymphedema.**

**Keywords** Apoptosis; Dietary Intervention; Endoplasmic Reticulum Stress; Fatty Acid-binding Protein 4; Oxidative Stress
**Subject Categories** Metabolism; Vascular Biology & Angiogenesis

## Introduction

Lymphedema is a chronic condition characterized by the accumulation of lymph fluid in the interstitial space, resulting from an insufficient or impaired function of the lymphatic system, which leads to tissue swelling, inflammation, and increased susceptibility to infections (Grada and Phillips, 2017). Secondary lymphedema is caused by an injury to the lymphatic system due to surgery, cancer treatments (radiation and/or chemotherapy), infections, trauma, and/or obesity (Koelmeyer et al, 2022; Stout et al, 2024). Increasing evidence suggests that secondary lymphedema develops through a multi-step process involving the role of events beyond the initial lymphatic insult in disease progression (Bowman and Rockson, 2024; Sung et al, 2022). The underlying reasons why individuals develop secondary lymphedema remain poorly understood.

Several risk factors contribute to the development of secondary lymphedema, including high body mass index (BMI), surgical lymph node removal, and extensive cancer treatments (Garza et al, 2017; Jammallo et al, 2013; Leray et al, 2020). In cancer patients who exhibit these risk factors, the incidence of lymphedema can be as high as 30–50% (Cemal et al, 2011; Shen et al, 2022). Despite these known associations, the mechanisms linking these risk factors to an increased incidence of lymphedema remain unclear.

The microenvironment of lymphedema is markedly altered, with elevated inflammation mediated by T cells and macrophages, which exacerbates disease progression (Brown et al, 2023). Additionally, there is an increase in vascular endothelial growth factor C (VEGF-C) in lymphedema tissue, primarily secreted by macrophages. Despite the presence of VEGF-C, lymphangiogenesis is impaired and dysfunctional (Gousopoulos et al, 2017b; Rutkowski et al, 2006). Areas of oxidative stress are also evident in the lymphedema microenvironment, and when combined with elevated VEGF-C levels, increased cell death in lymphatic endothelial cells (LECs) further prevents functional lymphangiogenesis (Hossain et al, 2024). These pathological conditions underscore the complexity of the lymphedema microenvironment, and the limited therapeutic approaches targeting the lymphedema microenvironment.

Lipotoxicity, a form of metabolic syndrome characterized by the accumulation of lipids in non-adipose tissues, plays a central role in various diseases, including cardiovascular diseases, diabetes, and obesity (Mann et al, 2024; Yazici and Sezer, 2017). In a healthy metabolic state, there is a balance between lipid synthesis, oxidation, and cellular uptake. With conditions such as obesity, diabetes, and heart disease, an overabundance of saturated fatty acids (SFAs) contributes to tissue damage through lipotoxicity

[1]Department of Oncology, University of Alberta, Edmonton, Alberta, Canada. [2]Department of Agricultural, Food and Nutritional Science, University of Alberta, Edmonton, Alberta, Canada. ✉E-mail: sgibson2@ualberta.ca

(Nolan and Larter, 2009; Piccolis et al, 2019). Obesity, characterized as a BMI of 30 or higher, is a well-established risk factor for lymphedema, and elevated SFA levels in the blood are commonly observed in obese individuals (Zhou et al, 2020). Furthermore, high fat diets (HFDs), which are typically rich in SFAs, have been shown to exacerbate lymphedema severity in animal models. Specifically, mice fed with HFDs display severe tail and hindlimb lymphedema (Blum et al, 2014; Khan et al, 2022; Savetsky et al, 2014). The role lipotoxicity plays in lymphedema progression is unknown.

In this study, we demonstrate that patients with secondary lymphedema exhibit a significantly lower PUFA/SFA ratio in plasma compared to non-lymphedema controls, reflecting an altered fatty acid composition. We also show that stearic acid (SA), a SFA, selectively induces cell death in LECs through endoplasmic reticulum (ER) stress. Furthermore, a high saturated fat diet (HSFD) in mice exacerbated tail lymphedema by prolonging ER stress and promoting LEC death. These findings suggest that changes in the relative proportions of PUFAs and SFAs may contribute to the pathophysiology of lymphedema.

## Results

### Patients with lymphedema have a lower PUFA/SFA ratio than non-lymphedema controls

Lipids, including fatty acids and lipid mediators, play a critical role in inflammatory responses, tissue remodeling, and cellular homeostasis. To evaluate the impact of circulating lipid levels on lymphedema, we measured the relative levels of fatty acids in 57 patients with secondary lymphedema compared to 22 non-lymphedema controls. Most of these patients (89%) were female and ranged in age from 32 to 88 years. Approximately 67% had a history of breast cancer-related lymphedema, 24% developed lymphedema due to other types of cancer, and 9% had non-cancer-related lymphedema. In addition, 72% of the patients presented with upper extremity lymphedema, the majority of which were cases of breast cancer-related lymphedema. The remaining 28% had lower extremity lymphedema, primarily associated with chronic venous insufficiency, gynecological cancers, prostate cancer, melanoma, and bladder cancer. The demographic characteristics of patients with lymphedema and non-lymphedema controls in our study are presented in Appendix Table S1. The plasma PUFA/SFA ratio in patients with lymphedema was significantly lower than non-lymphedema controls across different lipid fractions, including phospholipids (PLs; lymphedema: 0.86 vs. non-lymphedema controls: 0.93), triglycerides (TGs; lymphedema: 0.60 vs. non-lymphedema controls: 0.78), and cholesteryl esters (CEs; lymphedema: 4.04 vs. non-lymphedema controls: 4.59) (Fig. 1A–C). There was no significant difference observed in the PUFA/SFA ratio of free fatty acids (FFAs; lymphedema: 0.56 vs. non-lymphedema controls: 0.64) fraction (Fig. 1D). The non-lymphedema control group included both healthy individuals and breast cancer patients who did not develop lymphedema. The latter were monitored for up to 5 years following cancer treatment and remained lymphedema-free throughout the follow-up period. Fatty acid profiles from healthy controls were consistent with previously reported reference values (Larsen et al, 2018). Breast cancer controls showed similar levels to healthy controls except for FFAs, where the PUFA/SFA ratio was lower than healthy controls (Appendix Fig. S1). Furthermore, the low PUFA/SFA ratio in the patients with secondary lymphedema did not correlate with body

mass index (BMI) (Fig. 1E–H), indicating that the observed fatty acid imbalance is likely not explained by BMI. In contrast, there was a negative correlation between non-lymphedema controls and BMI in the TGs PUFA/SFA ratio (Appendix Fig. S2). The reduced PUFA/SFA ratio was consistent regardless of whether patients were diagnosed with breast cancer-related lymphedema or non-breast cancer-related lymphedema (Appendix Fig. S1), and it was notably more pronounced in female patients than males; however, the interpretation of this sex-based difference is limited by the relatively small number of male subjects included in the cohort (Appendix Fig. S3). In addition to a significant increase in SFAs in PLs and TGs, and a reduction in PUFAs in TGs and CEs in plasma from patients with lymphedema compared to controls (Fig. EV1A–D), analysis of individual fatty acid species indicates that the lower PUFA/SFA ratio in the lymphedema group was primarily driven by a higher proportion of palmitic acid (PA, 16:0) in the PL fraction and reduced levels of the omega-6 PUFA linoleic acid (LA, 18:2n6) across all four lipid fractions (Fig. EV1E–H; Appendix Fig. S4). These findings suggest that an altered fatty acid composition may play a role in the pathophysiology of lymphedema.

### High saturated fat diet prolongs swelling and induces ER stress in a tail lymphedema mouse model

To better understand the role of a low PUFA/SFA ratio in lymphedema development, we performed dietary interventions in a murine model of surgically induced tail lymphedema. Since most patients with lymphedema in our study were female and exhibited a greater reduction in the PUFA/SFA ratio, female C57BL/6J mice were randomly assigned to standard chow diet (CD), HFD, or HSFD (Appendix Fig. S5A). The composition of the diets used in this study is detailed in Appendix Table S2. Although statistically significant, a modest increase in body weight (~16%) was observed in the mice from the HFD and HSFD groups compared to the CD group (Appendix Fig. S5B). This level of weight gain in female C57BL/6 mice is insufficient to induce obesity-related metabolic changes, as this strain and sex are known to be resistant to diet-induced obesity over moderate durations (Casimiro et al, 2021).

After 4 weeks on their respective diets, tail lymphedema was surgically induced, and its progression was monitored for an additional 4 weeks. Mice fed with the CD and subjected to surgical lymphatic damage exhibited a progressive increase in tail swelling over a 22-day period compared to sham controls, followed by a reduction in swelling. Mice with lymphedema from both HFD and HSFD groups exhibited an earlier onset of tail swelling compared to the CD-fed lymphedema mice. The HSFD-fed mice showed a significant delay in the reduction in tail swelling, with tail volume remaining 62% higher 28 days post-surgery compared to both the HFD and CD groups (Fig. 2A–C). Only mice fed the HSFD displayed a significantly reduced PUFA/SFA ratio in their plasma compared to the CD group, reflecting the pattern observed in patients with lymphedema (Fig. EV2A). This was associated with an increased portion of the SFA SA (18:0) and a reduced portion of the omega-6 PUFA linoleic acid (18:2n6) in the mice plasma (Fig. EV2B).

To determine whether diets altered microenvironmental stresses in the tail lymphedema model, immunofluorescence staining was performed on tail sections collected 28 days post-lymphatic surgery. We evaluated ER stress markers spliced X-box binding protein 1 (sXBP-1) and C/EBP homologous protein (CHOP) in mice with lymphedema fed the HSFD, as lipotoxicity can induce ER stress (Han and Kaufman, 2016; Larsen et al, 2018). This revealed

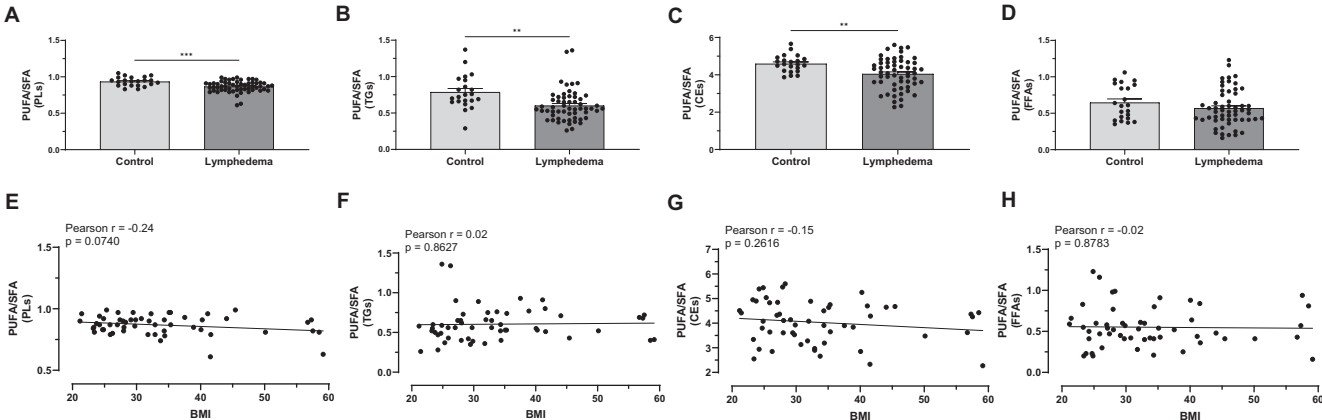

**Figure 1.   Reduced PUFA/SFA ratio in patients with secondary lymphedema compared to non-lymphedema controls.**

Plasma samples from patients with secondary lymphedema ($n = 57$) and non-lymphedema controls ($n = 22$) were analyzed to assess the polyunsaturated fatty acid (PUFA) to saturated fatty acid (SFA) ratio across different lipid fractions. Scatter plots showing individual PUFA/SFA ratios for (**A**) phospholipids (PLs), (**B**) triglycerides (TGs), (**C**) cholesteryl esters (CEs), and (**D**) free fatty acids (FFAs). Pearson correlation analysis indicates no significant relationship between the PUFA/SFA ratio of (**E**) PLs, (**F**) TGs, (**G**) CEs and (**H**) FFAs and body mass index (BMI) in patients with lymphedema. Data are presented as mean ± SEM. Statistical analysis: two-tailed unpaired $t$ test. Significance: *$P < 0.05$, ***$P < 0.001$. Exact $P$ values for the statistical comparisons are shown in Appendix Table S3. Source data are available online for this figure.

elevated levels of sXBP-1 and CHOP, accompanied by a marked increase in cell death, as indicated by terminal deoxynucleotidyl transferase dUTP nick end labeling (TUNEL) staining. These changes were observed both within the lymphatic vasculature (LYVE-1-positive cells), with sXBP-1, CHOP, and TUNEL levels increasing by threefold, fourfold, and threefold, respectively, and in the surrounding perilymphatic tissue, where the same markers increased by fivefold, fourfold, and twofold, respectively. Although these markers were also elevated in the HFD group (Fig. EV2C–I), the increases were notably less pronounced compared to the HSFD group (Fig. 2D–J). The presence of ER stress and cell death in the lymphedematous tissue of mice fed with the HSFD suggests that a diet high in saturated fat leads to a reduced PUFA/SFA ratio and prolonged swelling of tail lymphedema in this mouse model.

## Stearic acid selectively induced apoptosis in lymphatic endothelial cells through ER stress and reactive oxygen species

To investigate the in vitro effects of FFAs on lymphatic and vascular endothelial cells, human dermal lymphatic endothelial cells (HDLECs), human umbilical vein endothelial cells (HUVECs), and human dermal microvascular endothelial cells (HDMECs) were exposed to increasing concentrations of several different fatty acids, and cell death was evaluated using a trypan blue exclusion assay. Increasing concentrations of the SFAs SA and PA resulted in dose-dependent cell death in HDLECs, to a greater extent than HUVECs and HDMECs, indicating that HDLECs are more susceptible to lipotoxicity. Specifically, 24-h treatment with SA resulted in the following percentages of cell death: Vehicle: 10.2%, 1 µM: 15.5%, 10 µM: 26.8%, 50 µM: 58.0%, and 100 µM: 76.5% (Fig. 3A). In comparison, SFA-induced lipotoxicity was lower in HUVECs and HDMECs than in HDLECs, with HDLECs exhibiting a twofold increase in cell death with SA and a 1.3-fold increase with PA (Fig. 3B; Appendix Fig. S6). These findings suggest that LECs

are more susceptible to SFA-induced lipotoxicity than vascular endothelial cells. SA also induced dose-dependent cell death in rat mesenteric lymphatic endothelial cells (RMLECs) (Appendix Fig. S6E). To further confirm that SA induces apoptosis in LECs, we performed flow cytometry analysis with Annexin V/7-AAD staining. Treatment with SA led to a dose-dependent increase in the percentage of apoptotic cells, with 8% cell death observed in the vehicle-treated group, against 15% with 1 µM SA and 25% with 10 µM SA. These findings further underscore the role of SA in inducing apoptosis in HDLECs (Fig. 3C).

Corroborating the cell death data, SA selectively inhibited HDLECs growth while exerting minimal effects on HUVECs, as demonstrated by the clonogenic assay. HDLECs exhibited high sensitivity to SA, with 500 nM reducing the number of colonies from 269 to 52 (Appendix Fig. S7A,B). In contrast, the same concentration of SA inhibited colony growth in HUVECs from 105 to 75 (Appendix Fig. S7C,D). RMLECs required higher SA concentrations to observe significant inhibition of colony formation, with 1 µM reducing it by 1.3-fold and 10 µM reducing it by 37-fold (Appendix Fig. S7E,F). The lipid droplet formation detected by the Oil Red O staining was not different between HDLECs and HUVECs following SA treatment, suggesting a similar uptake of SA by these two cell lines (Appendix Fig. S8).

Given the high sensitivity of HDLECs to SA-induced cell death, we further investigated the mechanisms underlying these lipotoxic effects. SFAs are known to induce lipotoxicity through possible mechanisms involving oxidative stress and ER stress. While these mechanisms have been well-documented in various cell types, their role in LECs remains unknown. To address this, we investigated the involvement of oxidative and ER stress in SA-induced lipotoxicity in LECs. HDLECs were treated with 1 µM and 10 µM SA for 24 h and subsequently stained with MitoSOX Red and dihydroethidium (DHE) to detect mitochondrial and intracellular reactive oxygen species (ROS) levels by flow cytometry, respectively. SA treatment significantly increased ROS production in HDLECs, with mitochondrial ROS levels, measured by

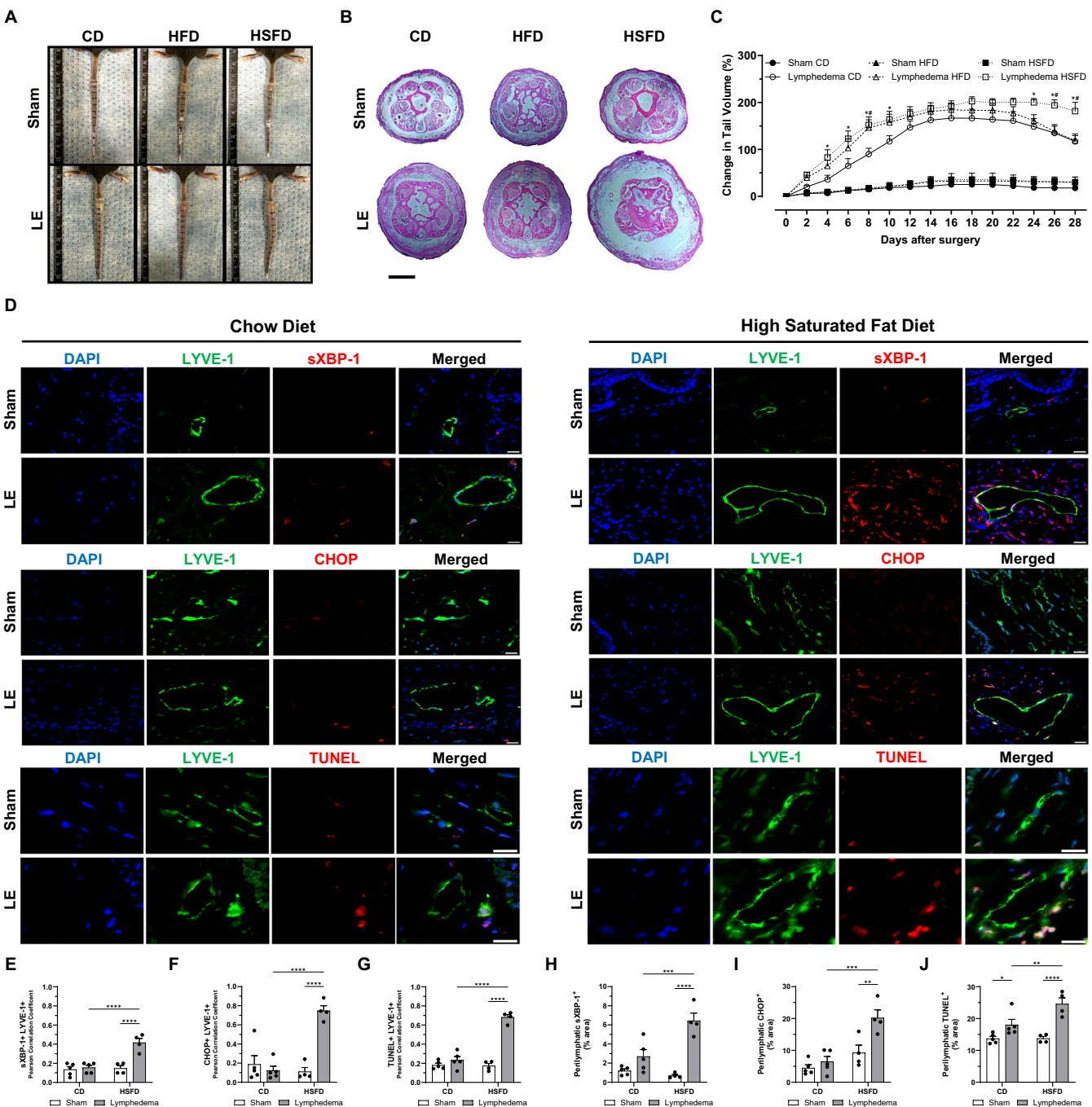

MitoSOX, elevated by sixfold, and intracellular ROS levels, assessed by DHE staining, elevated by fivefold (Fig. 3D,E; Appendix Fig. S9). In contrast, SA treatment did not significantly alter ROS levels in HDMECs (Appendix Fig. S10). To confirm the role of oxidative stress, we pretreated HDLECs with the antioxidant α-tocopherol 1 h before SA exposure. The pretreatment with α-tocopherol significantly reduced SA-induced cell death by 46%, further implicating oxidative stress in SA-mediated lipotoxicity in LECs (Fig. 3F). In addition, we examined the involvement of ER stress, a critical pathway in SFA-induced lipotoxicity, by evaluating protein levels of sXBP-1 and CHOP. Western blot analysis revealed a dose-dependent increase in

sXBP-1 and CHOP expression following SA treatment (Appendix Fig. S11), suggesting that ER stress also contributes to SA-induced cell death in LECs. These findings demonstrate that ROS and ER stress contribute to SA-induced cell death in LECs.

## Stearic acid inhibits cell migration and vessel sprouting in lymphatic endothelial cells

Following lymphatic injury, lymphangiogenesis is initiated, during which LECs migrate, proliferate, and form new vessels to restore lymphatic function. To assess whether SA impairs this process, we

◄

**Figure 2. High saturated fat diet induces endoplasmic reticulum (ER) stress and prolongs tail lymphedema in a mouse model.**

Lymphedema (LE) was induced via lymphatic ablation in the tails of mice maintained on a standard chow diet (CD), a high-fat diet (HFD), or a high-saturated fat diet (HSFD). Sham surgery involved only skin incision. (A) Representative images of mouse tails from each dietary group at day 28 post-surgery. (B) Representative H&E staining of tail tissues from sham and lymphedema mice across dietary groups, illustrating differences in tail swelling. Scale bars: 500 µm. (C) Tail volume measurements over time, showing significantly prolonged swelling in HSFD-fed mice compared to CD and HFD groups ($n = 5$ mice). The lymphedema volume curve for CD-fed mice peaks at days 16–22, followed by spontaneous resolution. (D) Representative immunofluorescence staining of LYVE-1-positive cells (green) and markers of ER stress (sXBP-1, CHOP) or apoptosis (TUNEL) (red) in tail tissues from sham or lymphedema mice across diets, 28 days post-surgery. DAPI stains the nucleus (blue). Scale bars: 50 µm. Scatter plots showing (E, H) sXBP-1 + LYVE-1+ colocalization and perilymphatic sXBP-1 intensities, (F, I) CHOP + LYVE-1+ colocalization and perilymphatic CHOP intensities, and (G, J) TUNEL + LYVE-1+ colocalization and perilymphatic TUNEL intensities comparing the groups ($n = 4$–5 mice). Data are presented as mean ± SEM. Statistical analysis: two-way ANOVA with Šídák's post hoc test for (C). Significance: $*P < 0.05$ Lymphedema CD vs Lymphedema HSFD, $^\#P < 0.05$ Lymphedema HFD vs Lymphedema HSFD; one-way ANOVA with Tukey's post hoc test for (E–J). Significance: $*P < 0.05$, $**P < 0.01$, $***P < 0.001$, $****P < 0.0001$. Exact $P$ values for the statistical comparisons are shown in Appendix Table S3. LYVE-1 lymphatic vessel endothelial hyaluronan receptor-1, sXBP-1 spliced X-box binding protein 1, CHOP C/EBP homologous protein, TUNEL terminal deoxynucleotidyl transferase dUTP nick end labeling, DAPI 4′,6-diamidino-2-phenylindole. Source data are available online for this figure.

performed a scratch assay to evaluate LEC migration. Under control conditions, HDLECs significantly closed the wound area within 32 h. However, in the presence of 10 µM SA, wound closure was markedly impaired, indicating a disruption of LEC migratory capacity (Fig. EV3A,B). In contrast, HDMECs were able to significantly reduce the wound area within 48 h, regardless of the presence of 10 µM SA, indicating that their migratory capacity was not affected (Fig. EV3C,D). To further investigate the impact of SA on lymphangiogenesis, we performed a lymphatic ring assay using thoracic lymphatic trunks excised from mice. Rings were sectioned and embedded in Matrigel supplemented with growth medium. In control conditions, lymphatic vessels exhibited robust cellular proliferation, migration, and new vessel sprouting (red arrows). In contrast, rings treated with 10 µM SA showed no evidence of these lymphangiogenic processes (Fig. EV3E). Together, these findings suggest that SA impairs critical steps in lymphangiogenesis, including endothelial proliferation, migration, and sprouting.

## Oxidative stress potentiates stearic acid-induced apoptotic cell death in lymphatic endothelial cells

Our previous study demonstrated a significant increase in 8-hydroxy-2′-deoxyguanosine (8-OHdG), a marker of oxidative DNA damage, 12 days after lymphatic ablation in a mouse model of tail lymphedema (Hossain et al, 2024). Building on these findings, we evaluated the impact of dietary interventions on oxidative stress markers in lymphedema. Immunofluorescence analysis demonstrated that 28 days post-lymphatic injury, mice with lymphedema fed the HSFD exhibited significantly higher levels of 8-OHdG, with a Pearson correlation of 0.54 with LYVE-1 positive cells, compared to the CD- (Pearson correlation: 0.21) and the HFD- (Pearson correlation: 0.31) groups (Fig. 4A,B; Appendix Fig. S12).

To further investigate the role of oxidative stress in SA-induced lipotoxicity, HDLECs were pre-treated with 1 µM or 10 µM SA for 1 h, followed by exposure to 300 µM hydrogen peroxide ($H_2O_2$) for 24 h. We found that combined SA and $H_2O_2$ treatment significantly increased cell death by twofold when compared to either treatment alone, as indicated by Annexin V/7-AAD assay (Fig. 4C; Appendix Fig. S9A). Oxidative stress, assessed by flow cytometry via MitoSOX Red and DHE staining, also showed significantly elevated mitochondrial and intracellular ROS levels in cells treated with both SA and $H_2O_2$ (Fig. 4D,E; Appendix Fig. S9B,C). Western blot analysis was performed to elucidate the mechanisms driving the

SA-induced lipotoxicity. HDLECs treated with SA and $H_2O_2$ showed reduced expression of the anti-apoptotic proteins Bcl-2 and Mcl-1, along with decreased levels of total caspase-3 and total PARP-1, indicating apoptosis activation (Appendix Fig. S13). These findings indicate that oxidative stress potentiates the SA induced apoptosis in LECs and suggest a mechanistic link between dietary saturated fat, oxidative stress, and lymphatic endothelial dysfunction in lymphedema.

The omega-6 PUFA LA can act as an antioxidant. Given the increased oxidative stress and reduced LA proportions in plasma, we investigated whether LA could mitigate SA-induced lipotoxicity. HDLECs were pretreated with 50 µM LA for 30 min before a 24-h exposure to 50 µM SA. LA pretreatment significantly protected cells from SA-induced cytotoxicity: while SA alone caused ~44% cell death, LA pretreatment reduced cell death to 19% (Appendix Fig. S14). These results suggest that LA effectively counteracts SA-induced lipotoxic effects in HDLECs.

## Dietary change prevents prolonged tail lymphedema induced by a high saturated fat diet in a mouse model

To evaluate whether the detrimental effects of HSFD on lymphedema progression could be ameliorated, we transitioned mice from the HSFD to the CD 6 days after surgical lymphedema ablation. This dietary intervention resulted in a significantly less tail swelling, with the HSFD → CD group showing a 30% decrease in tail volume compared to the HSFD group by day 28 post-surgery (Fig. 5A,B). Despite exhibiting a greater increase in tail volume during the early phases of lymphedema progression when compared to mice with lymphedema maintained on CD, the tail volume of mice transitioned from HSFD to CD 6 days after surgery became comparable to that of CD-fed lymphedema mice by 28 days post-surgery. This indicates that transitioning to a CD effectively prevents the sustained tail swelling associated with HSFD consumption.

Furthermore, mice with lymphedema transitioned to the CD after HSFD showed significantly lower levels of oxidative and ER stress. This was evidenced by reduced expression of the oxidative stress marker 8-OHdG, and the ER stress markers sXBP1 and CHOP in both LECs (LYVE-1 positive) and the perilymphatic tissue at 28 days post-surgery. Similarly, cell death, as indicated by TUNEL staining, was also significantly less in both compartments (Fig. 5C–I; Appendix Fig. S15). Thus, switching from an HSFD to a CD after the onset of lymphedema effectively prevented tail

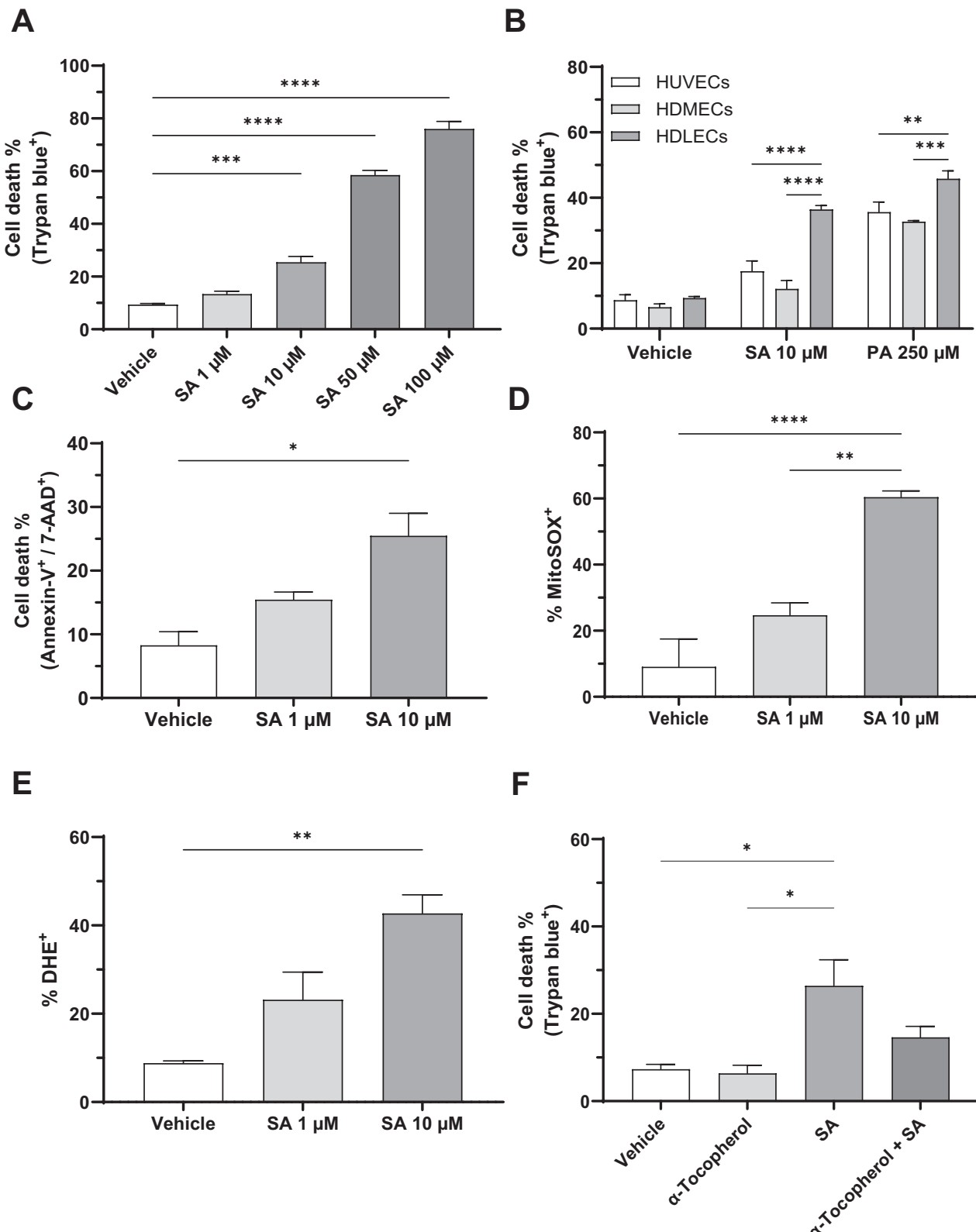

**Figure 3. Stearic acid promotes apoptosis in lymphatic endothelial cells via reactive oxygen species (ROS).**

(A) Human dermal lymphatic endothelial cells (HDLECs) were treated with increasing concentrations of stearic acid (SA) for 24 h ($n = 3$ independent experiments). Cell viability was assessed by trypan blue exclusion assay. (B) Human Umbilical Vein Endothelial Cells (HUVECs), Human Dermal Microvascular Endothelial Cells (HDMECs), and HDLECs were treated with 10 μM of SA and 250 μM of palmitic acid (PA) for 48 h ($n = 3$ independent experiments). Cell viability was measured by trypan blue exclusion assay. (C) Quantification of annexin V/7-AAD-positive HDLECs after 24 h of SA treatment ($n = 3$ independent experiments). (D) Quantification of MitoSOX-positive HDLECs, indicating mitochondrial ROS levels following SA treatment for 24 h ($n = 3$ independent experiments). (E) Quantification of DHE-positive HDLECs, measuring cytosolic ROS levels after 24 h of SA treatment ($n = 3$ independent experiments). (F) HDLECs pretreated with α-tocopherol (200 μg/mL) 1 h before SA treatment, with cell viability evaluated by trypan blue assay 24 h later ($n = 3$ independent experiments). Data are presented as mean ± SEM. Statistical analysis: one-way ANOVA with Tukey's post hoc test. Significance: *$P < 0.05$, **$P < 0.01$, ***$P < 0.001$, ****$P < 0.0001$. Exact $P$ values for the statistical comparisons are shown in Appendix Table S3. MitoSOX mitochondrial superoxide, DHE dihydroethidium. Source data are available online for this figure.

swelling and reduced key markers of cellular stress, including oxidative stress, ER stress, and cell death.

## FABP4 inhibition attenuates stearic acid-induced lipotoxicity in lymphatic endothelial cells

To further investigate the mechanism underlying the sensitivity of LECs to lipotoxicity, we investigated whether fatty acid transporting proteins contribute to the observed lipotoxic effects. Fatty acid-binding protein 4 (FABP4) is a key lipid chaperone that regulates intracellular fatty acid trafficking and metabolism. FABP4 levels are elevated in obesity and cancer, and mutations in the gene have been identified in cases of primary lymphedema (Ferrell et al, 2008; Zeng et al, 2020). We first measured circulating FABP4 levels in plasma samples from patients with lymphedema and non-lymphedema controls. FABP4 levels were 2.8-fold higher in individuals with lymphedema compared to controls (Fig. 6A; Appendix Fig. S16). These results suggest a potential association between elevated FABP4 levels and the presence of lymphedema.

To explore the role of FABP4 in SFA-induced lipotoxicity in LECs, we performed western blot analysis and found that HDLECs express significantly higher levels of FABP4 protein compared to HUVECs and HDMECs (Appendix Fig. S17A). To further evaluate the functional significance of FABP4 in SFA-induced lipotoxicity in vitro, we pretreated HDLECs with BMS-309403 (BMS), a selective FABP4 inhibitor, 30 min before a 24-h exposure to SA or PA. A concentration of 5 μM BMS was selected for use, as this was identified as the lowest dose that does not exhibit toxic effects in HDLECs (Appendix Fig. S17B). FABP4 inhibition significantly reduced cell death induced by both SFAs by ~50% (Fig. 6B). Furthermore, western blot analysis showed that BMS treatment markedly reduced the protein levels of the ER stress markers sXBP1 and CHOP in SA-treated HDLECs (Fig. 6C).

Immunofluorescence staining for FABP4 and the ER marker calreticulin revealed colocalization in HDLECs under control conditions (Pearson's correlation coefficient = 0.60), indicating ER localization of FABP4 in LECs. However, the 24-h exposure to SA or FABP4 inhibition did not significantly changed this colocalization pattern (Fig. 6D,E). To validate that the effects of BMS were specifically mediated through FABP4, we performed siRNA-mediated knockdown of FABP4 in HDLECs, which significantly decreased SA-induced cell death (Fig. 6F,G).

To assess lipid accumulation in lymphedematous tissue, we performed Oil Red O staining. Mice fed either a HFD or a HSFD exhibited increased lipid deposition in the lymphedematous region compared to those on the standard chow diet (Appendix Fig. S18).

Pharmacological inhibition of FABP4 with BMS markedly reduced lipid accumulation in HSFD-fed lymphedema mice, reaching levels comparable to those observed in sham-operated or CD-switched animals (Fig. EV4A,B). Immunostaining for FABP4 further revealed an increased number and size of adipocytes in lymphedematous tissue, both of which were significantly reduced following BMS treatment and dietary transition (Fig. EV4C–E). Additionally, HSFD feeding significantly increased circulating FABP4 levels in mice with lymphedema compared to those fed a CD, whereas FABP4 inhibition with BMS attenuated this elevation in HSFD-fed animals (Fig. EV4F).

## FABP4 inhibition attenuates tail swelling and oxidative stress in tail lymphedema induced by high saturated fat diet

Since FABP4 inhibitor blocks SA-induced cell death in HDLECs, we determined whether HFSD alters FABP4 expression. Immunofluorescence staining of tail sections showed increased FABP4 levels in the perilymphatic tissue of HSFD-fed mice with lymphedema compared to both their sham controls and mice in the CD and HFD lymphedema groups. Although a fivefold increase in FABP4 expression was observed in the HFD-fed lymphedema group compared to the CD group, this increase was less pronounced than the 6.8-fold increase in the HSFD-fed group (Appendix Fig. S19). These findings suggest that HSFD consumption leads to a robust increase in perilymphatic FABP4 expression in lymphedema, potentially implicating it as a mediator of HSFD-induced lymphatic dysfunction.

To investigate the therapeutic potential of FABP4 inhibition in the HFSD-fed mouse tail lymphedema model, we treated CD- and HSFD-fed mice with the selective FABP4 inhibitor BMS. After 4 weeks of dietary intervention, daily oral BMS treatment was administered by gavage for an additional 4 weeks, starting on the day of surgical lymphedema induction. FABP4 inhibition significantly alleviated tail swelling in HSFD-fed mice, resulting in a 54% reduction in tail volume compared to the vehicle-treated HSFD group. This effect was absent in CD-fed mice, indicating that FABP4 inhibition specifically targets the exacerbating effects of HSFD on lymphedema progression (Fig. 7A,B). BMS treatment attenuated oxidative stress in HSFD-fed mice, as demonstrated by decreased levels of the oxidative stress marker 8-OHdG in both LYVE-1-positive LECs and the surrounding perilymphatic tissue, with reductions of 37% and 44%, respectively (Appendix Fig. S20). Additionally, BMS treatment mitigated ER stress, as evidenced by reduced expression of the ER stress markers sXBP1 and CHOP in

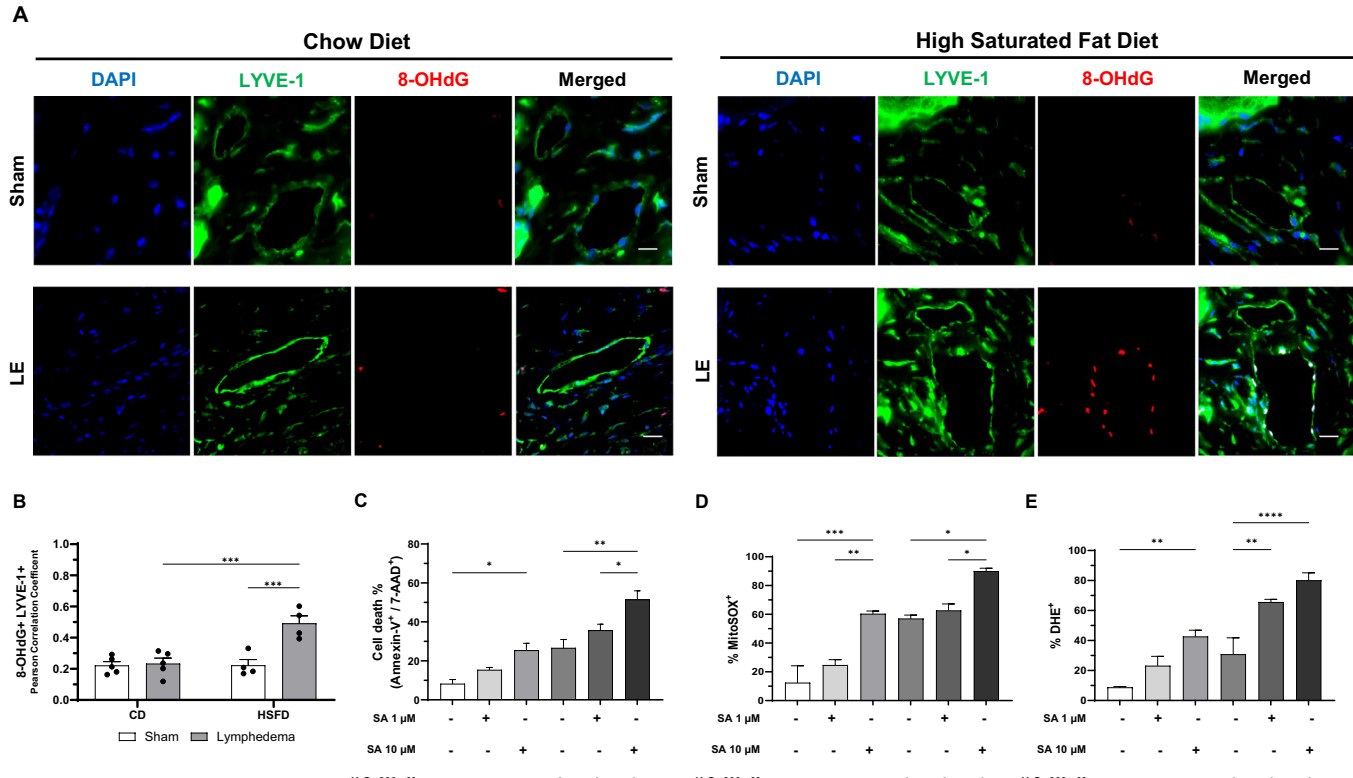

**Figure 4.  Oxidative stress potentiates stearic acid-induced apoptotic cell death in lymphatic endothelial cells.**

(A) Representative immunofluorescence staining of 8-OHdG (red), a marker of oxidative DNA damage, colocalized with LYVE-1-positive cells (green) in tail tissues of mice with lymphedema (LE) fed a chow diet (CD) or high saturated-fat diet (HSFD) 28 days post-surgery. DAPI stains the nucleus (blue). Scale bars: 50 μm. (B) Scatter plots showing 8-OHdG+ LYVE-1+ colocalization ($n = 4$–5 mice). (C) Quantification of annexin V/7-AAD-positive HDLECs pre-treated with 10 μM of stearic acid (SA) for 1 h, followed by 300 μM of hydrogen peroxide ($H_2O_2$) for 24 h ($n = 3$ independent experiments). (D, E) Quantification of mitochondrial and intracellular reactive oxygen species (ROS) using MitoSOX Red and DHE staining, respectively ($n = 3$-4 independent experiments). Data are presented as mean ± SEM. Statistical analysis: one-way ANOVA with Tukey's post hoc test for (B–E). Significance: *$P < 0.05$, **$P < 0.01$, ***$P < 0.001$, ****$P < 0.0001$. Exact $P$ values for the statistical comparisons are shown in Appendix Table S3. 8-OHdG 8-hydroxy-2'-deoxyguanosine, LYVE-1 lymphatic vessel endothelial hyaluronan receptor-1, DAPI 4',6-diamidino-2-phenylindole, MitoSOX mitochondrial superoxide, DHE dihydroethidium, Bcl-2 B-cell leukemia/lymphoma 2 protein, Mcl-1 myeloid cell leukemia sequence 1, PARP-1 poly(ADP-Ribose) polymerase 1. Source data are available online for this figure.

both LYVE-1-positive LECs and perilymphatic tissue, with decreases of 28% and 45% for sXBP1 (Fig. 7D,G; Appendix Fig. S21) and 62% and 36% for CHOP (Fig. 7E,H; Appendix Fig. S21), respectively. Furthermore, FABP4 inhibition significantly reduced cell death in LECs, as indicated by decreased TUNEL-positive staining in both LYVE-1-positive cells and the surrounding perilymphatic tissue, with reductions of 47% and 21%, respectively (Fig. 7F,I; Appendix Fig. S21). The protein levels detected in control mice 28 days post-sham surgery treated with vehicle (PBS) are shown in Appendix Fig. S22.

During lymphedema, infiltration of immune cells, particularly macrophages and CD4+ T cells, is increased in affected tissues. To assess whether FABP4 inhibition or dietary intervention modulates immune cell presence, we performed immunostaining for macrophages (F4/80) and CD4+ T cells in lymphedematous tails from mice fed a CD or a HSFD. HSFD-fed mice exhibited a marked increase in perilymphatic immune cell infiltration. F4/80+ macrophage area (Fig. EV5A,C) increased by ~2.8-fold compared to CD-fed controls, while CD4+ T cell area (Fig. EV5B,D) increased by about 2.2-fold. Treatment with the FABP4 inhibitor BMS-309403

significantly prevented this immune cell accumulation, bringing macrophage levels down by ~50% (Fig. EV5A,C) and CD4+ T cells down by ~40% (Fig. EV5B,D) compared to untreated HSFD mice. These findings indicate that FABP4 inhibition attenuates HSFD-induced immune cell recruitment around lymphatic vessels.

Collectively, these findings suggest that SFAs and FABP4 contribute to lymphedema progression by promoting ER stress and oxidative stress, ultimately leading to apoptosis of LECs.

## Discussion

Obesity is a well-established risk factor for lymphedema, with obese individuals undergoing cancer surgery or treatment exhibiting a significantly higher risk of developing this lymphatic disease (Garza et al, 2017; Jammallo et al, 2013; Leray et al, 2020; Shen et al, 2022). In cases of extreme obesity (BMI ≥40), lymphatic function is profoundly impaired, further contributing to lymphedema development (Arngrim et al, 2013; Greene et al, 2015). While some pre-clinical studies have demonstrated an association between HFDs and increased

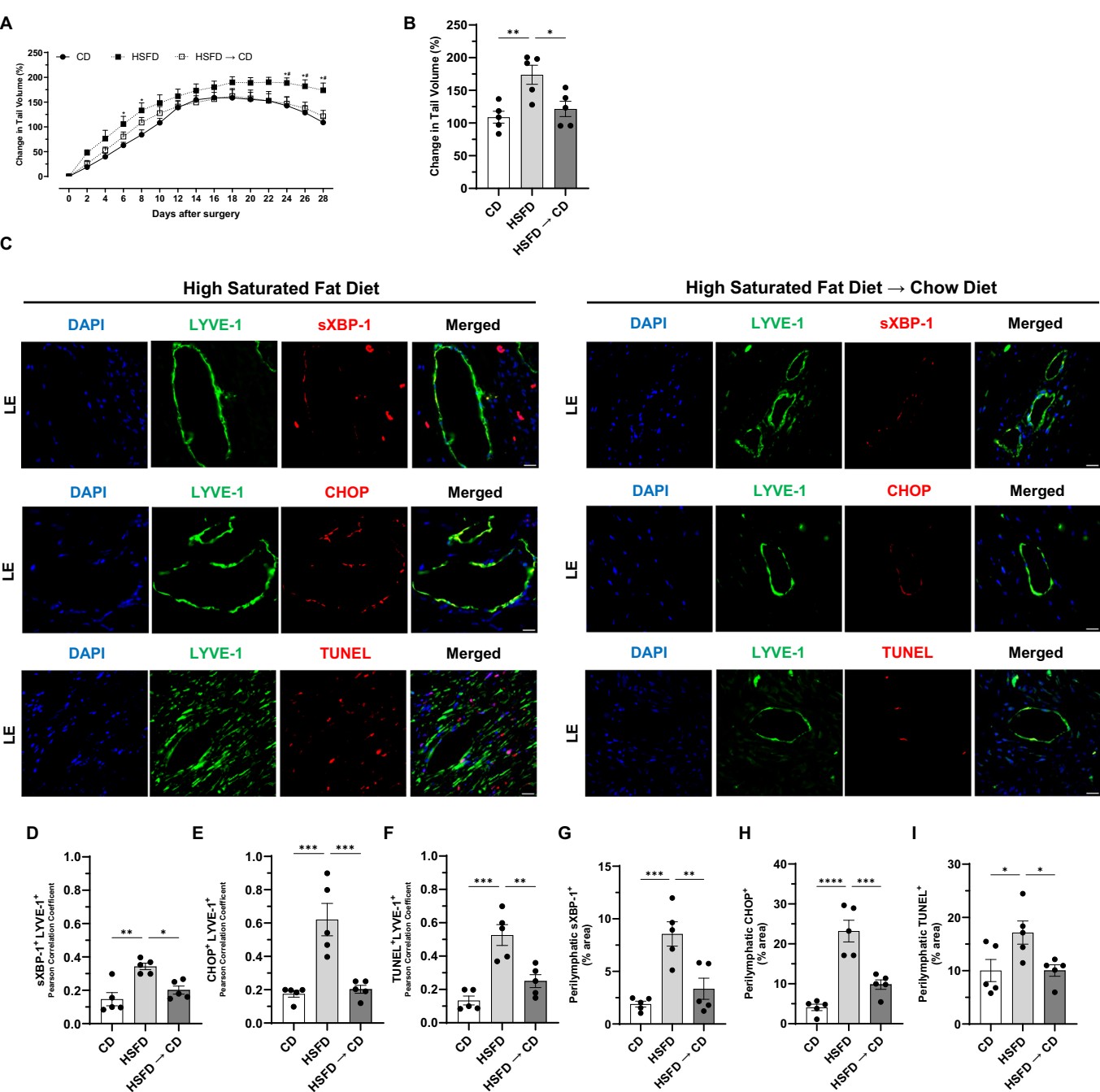

**Figure 5. Transition from high saturated fat diet (HSFD) to chow diet (CD) prevents prolonged tail lymphedema induced by HSFD in a mouse model.**

(**A**) Tail volume measurements of mice with lymphedema (LE) over time, demonstrating that the significantly prolonged swelling observed in HSFD-fed mice is prevented when the diet is transitioned from HSFD to CD (HSFD → CD) starting 6 days after surgical lymphatic ablation ($n = 5$ mice). (**B**) Scatter plots showing the changes in tail volume 28 days post lymphatic surgery ($n = 5$ mice). (**C**) Representative immunofluorescence staining of LYVE-1-positive cells (green) and markers of ER stress (sXBP-1, CHOP), and apoptosis (TUNEL) (red) in tail tissues from lymphedema mice across diets, 28 days post-surgery. DAPI stains the nucleus (blue). Scale bars: 50 μm. Scatter plots showing (**D**, **G**) sXBP-1 + LYVE-1+ colocalization and perilymphatic sXBP-1 intensities, (**E**, **H**) CHOP + LYVE-1+ colocalization and perilymphatic CHOP intensities, and (**F**, **I**) TUNEL + LYVE-1+ colocalization and perilymphatic TUNEL intensities comparing the groups ($n = 5$ mice). Data are presented as mean ± SEM. Statistical analysis: two-way ANOVA with Šídák's post hoc test for (**A**). Significance: *$P < 0.05$ Lymphedema HSFD vs Lymphedema CD, #$P < 0.05$ Lymphedema HSFD vs Lymphedema HSFD → CD; one-way ANOVA with Tukey's post hoc test for (**B**, **D–I**). Significance: *$P < 0.05$, **$P < 0.01$, ***$P < 0.001$, ****$P < 0.0001$. Exact $P$ values for the statistical comparisons are shown in Appendix Table S3. LYVE-1 lymphatic vessel endothelial hyaluronan receptor-1, sXBP-1 spliced X-box binding protein 1, CHOP C/EBP homologous protein, TUNEL terminal deoxynucleotidyl transferase dUTP nick end labeling, DAPI 4′,6-diamidino-2-phenylindole. Source data are available online for this figure.

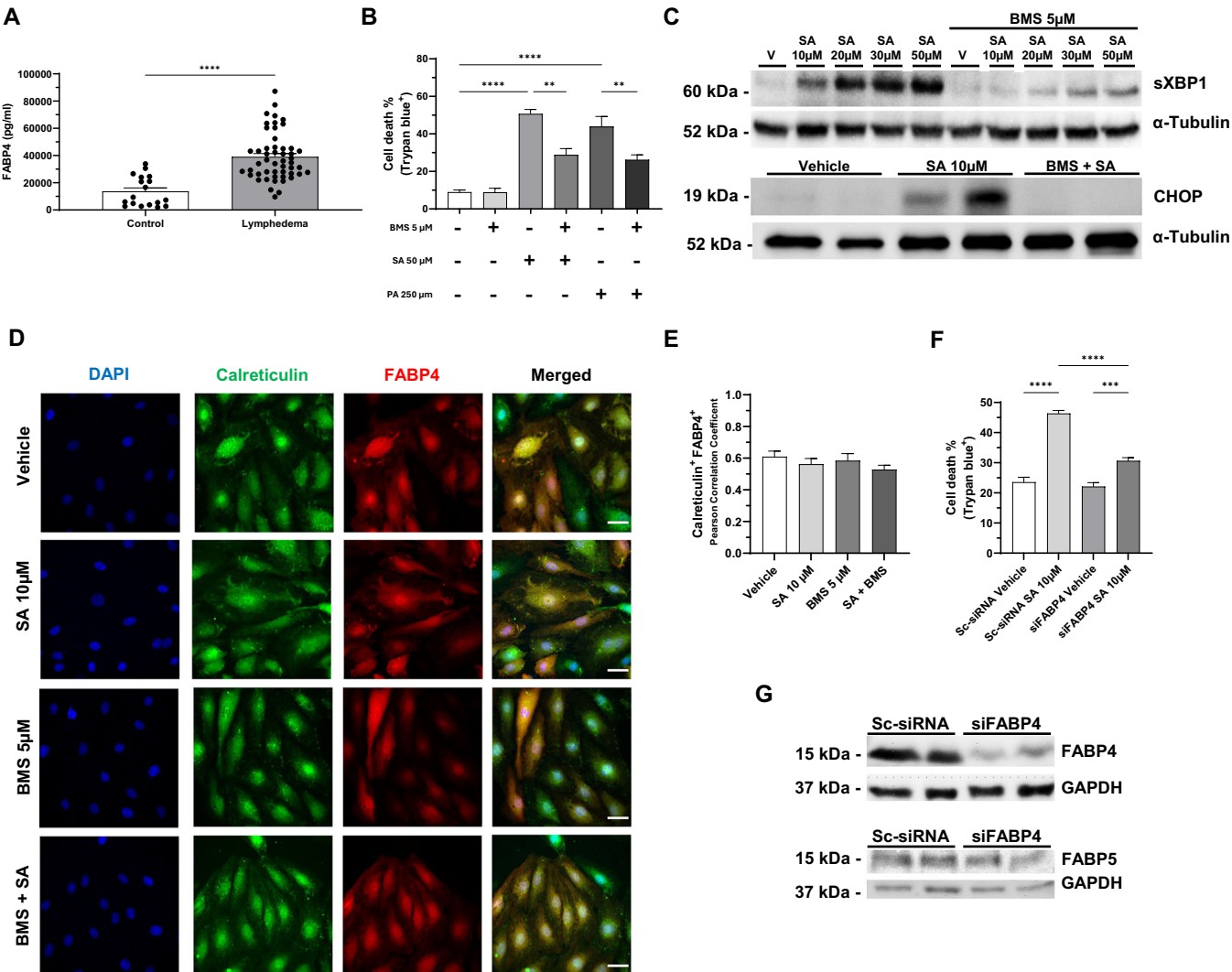

**Figure 6. Inhibition of FABP4 attenuates stearic acid-induced lipotoxicity and endoplasmic reticulum (ER) stress in lymphatic endothelial cells.**

(A) Circulating levels of fatty acid-binding protein 4 (FABP4) measured by ELISA in plasma samples from patients with lymphedema (n = 51) and non-lymphedema controls (n = 18). (B) Quantification of cell death (trypan blue assay) in HDLECs treated with SA (50 µM) or palmitic acid (PA, 250 µM) with or without the FABP4 inhibitor BMS-309403 (BMS, 5 µM) pre-treatment (n = 4 independent experiments). (C) Representative western blot of ER stress markers (sXBP1 and CHOP) in HDLECs treated with SA at increasing concentrations (10–50 µM) with or without BMS pre-treatment. (D) Representative immunofluorescence images of HDLECs treated with SA and BMS showing FABP4 (red) and the endoplasmic reticulum marker calreticulin (green). DAPI stains the nucleus (blue). Scale bars: 50 µm. (E) Colocalization quantification between FABP4 and calreticulin using Pearson's correlation coefficient, showing no significant differences between treatment groups (n = 3 independent experiments). (F) Quantification of cell death in HDLECs transfected with siFABP4 or scrambled siRNA (Sc-siRNA), with or without SA treatment, showing reduced cell death following FABP4 knockdown (n = 6 independent experiments). (G) Western blot confirming FABP4 knockdown in HDLECs transfected with FABP4-targeting siRNA (siFABP4), with no effect on FABP5 expression (n = 2). Data are presented as mean ± SEM. Statistical analysis: two-tailed unpaired t test for (A); one-way ANOVA with Tukey's post hoc test for (B, E, F). Significance: **P < 0.01, ***P < 0.001, ****P < 0.0001. Exact P values for the statistical comparisons are shown in Appendix Table S3. FABP4 fatty acid binding protein 1, sXBP-1 spliced X-box binding protein 1, CHOP C/EBP homologous protein, DAPI 4′,6-diamidino-2-phenylindole. Source data are available online for this figure.

lymphedema severity in both tail and hindlimb lymphedema models (Blum et al, 2014; Khan et al, 2022; Savetsky et al, 2014), others have reported that HFDs do not affect lymphedema in non-obese mice (Choi et al, 2023; Gousopoulos et al, 2017a). Furthermore, there is currently no evidence indicating that HFD increases the incidence of lymphedema. Our findings provide novel insights by demonstrating that both HFD and HSFD exacerbate tail lymphedema in the early stages of disease progression in mice. The lack of diet-induced obesity in our animal model is attributable to the use of female C57BL/6J mice,

which are more resistant to obesity under dietary stress, and the relatively short duration (8 weeks) of dietary intervention (Stapleton et al, 2024; Stubbins et al, 2012). Notably, only the HSFD delayed the spontaneous resolution of swelling in the mouse tail lymphedema model, indicating the unique impact of dietary SFA content. In addition, our study suggests that the progression of lymphedema may also be exacerbated by alterations in the proportion of PUFAs to SFAs in the lymphatic microenvironment. This is supported by our observation of a lower PUFA/SFA ratio in patients with lymphedema,

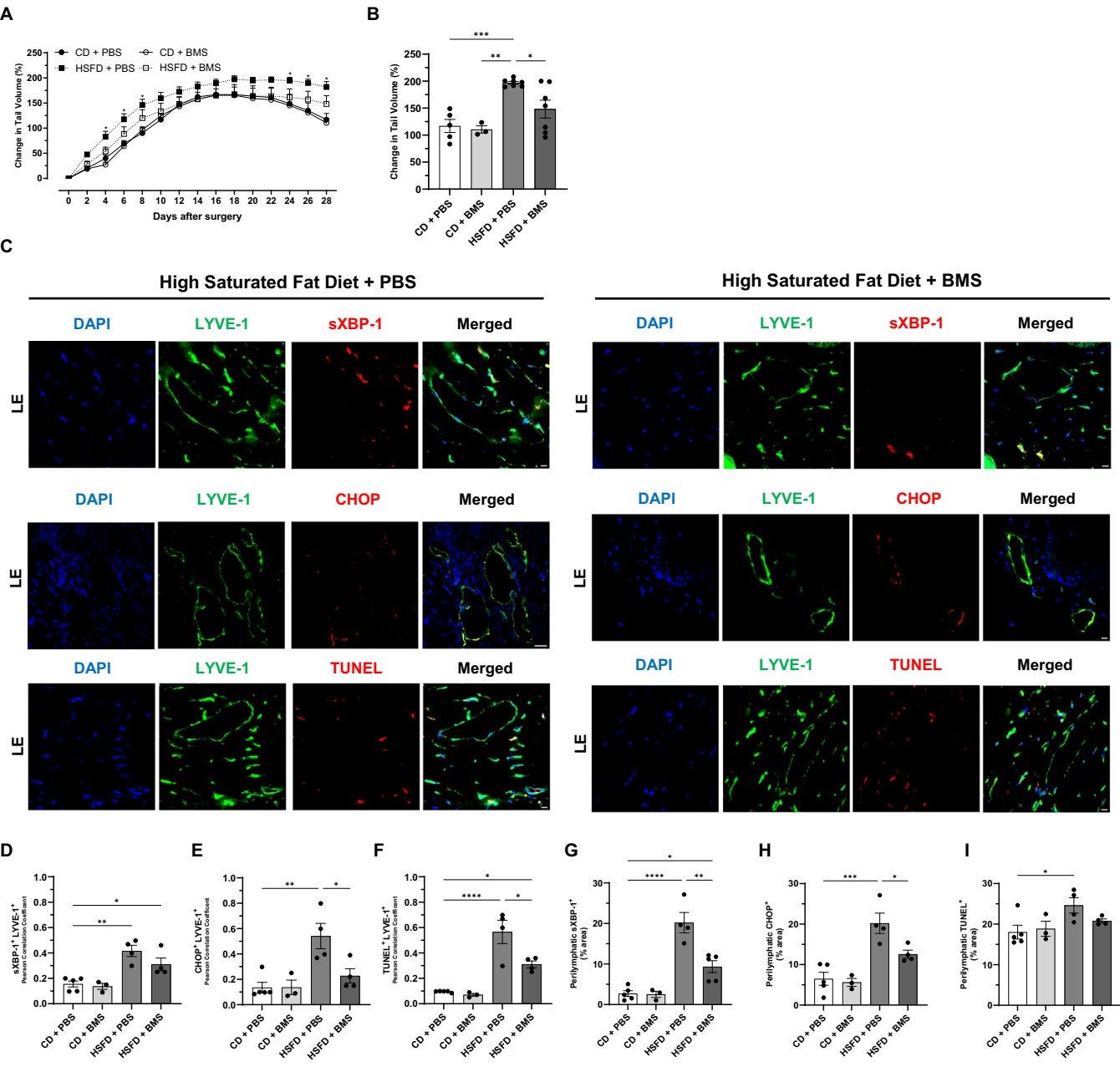

**Figure 7. FABP4 inhibition attenuates prolonged tail lymphedema induced by high saturated fat diet in a mouse model.**

(A) Tail volume measurements over time in mice with lymphedema (LE), showing that prolonged swelling induced by HSFD is significantly reduced with treatment using the FABP4 inhibitor BMS-309403 (HSFD + BMS) compared to HSFD + PBS controls ($n = 3$–7 mice). (B) Scatter plots showing the changes in tail volume 28 days post-lymphatic surgery ($n = 3$–7 mice). (C) Representative immunofluorescence staining of LYVE-1-positive cells (green) and markers of ER stress (sXBP-1, CHOP) and apoptosis (TUNEL) (red) in tail tissues from lymphedema mice across diets, 28 days post-surgery. DAPI stains the nucleus (blue). Scale bars: 25 μm. Scatter plots showing (D, G) sXBP-1 + LYVE-1+ colocalization and perilymphatic sXBP-1 intensities, (E, H) CHOP + LYVE-1+ colocalization and perilymphatic CHOP intensities, and (F, I) TUNEL + LYVE-1+ colocalization and perilymphatic TUNEL intensities comparing the groups ($n = 3$–7 mice). Data are presented as mean ± SEM. Statistical analysis: two-way ANOVA with Šídák's post hoc test for (A); one-way ANOVA with Tukey's post hoc test for (B, D–I). Significance: *$P < 0.05$, **$P < 0.01$, ***$P < 0.001$, ****$P < 0.0001$. Exact $P$ values for the statistical comparisons are shown in Appendix Table S3. LYVE-1 lymphatic vessel endothelial hyaluronan receptor-1, sXBP-1 spliced X-box binding protein 1, CHOP C/EBP homologous protein, TUNEL terminal deoxynucleotidyl transferase dUTP nick end labeling, DAPI 4′,6-diamidino-2-phenylindole. Source data are available online for this figure.

independent of BMI. These findings suggest that metabolic imbalances, particularly those involving dietary fatty acid composition, may play a key role in lymphedema pathogenesis.

LECs require fatty acid β-oxidation (FAO) for lymphangiogenesis, which involves proliferation, migration, and sprouting. FAO supports these processes by generating acetyl-coenzyme A (acetyl-CoA), a critical metabolite necessary for lymphatic differentiation (Wong et al, 2018). Alternate sources of acetyl-CoA, such as ketones, can rescue lymphangiogenesis under FAO inhibition and ketogenic diets have demonstrated benefits in pre-clinical

lymphedema models. These diets improved lymphatic vessel function and growth, reduced infiltration of anti-lymphangiogenic immune cells, and decreased tail swelling in lymphedema models (Garcia-Caballero et al, 2019). Similar benefits have been observed in patients, albeit without accounting for microenvironmental stressors (Lodewijckx et al, 2024). However, while fatty acid metabolism is essential for LEC function, not all lipid exposure is beneficial. An excess of SFAs may overwhelm the metabolic capacity of LECs, leading to lipotoxicity. In our study, SFAs such as SA induced apoptosis in LECs and contributed to sustained ER stress and oxidative stress in lymphedematous tissue. Switching from HSFD to a standard chow diet significantly prevented tail swelling, ER stress, and oxidative stress, underscoring the therapeutic potential of dietary modulation. Future studies will investigate the optimal timing of dietary intervention to prevent or mitigate lymphedema progression following lymphatic injury.

Lymphedema microenvironment is also altered by inflammation. Lipid metabolites contribute to inflammation through reduced specialized pro-resolving mediators (SPM) generated by the 15-lipoxygenase (15-LO) and increased leukotriene $LTB_4$ (Singh and Rao, 2019; Zamora et al, 2024). Human lipoxygenases catalyze the stereoselective dioxygenation of PUFAs, including arachidonic acid (AA), eicosapentaenoic acid (EPA) and DHA. Treatment with ketoprofen, a nonsteroidal anti-inflammatory drug that inhibits 5-lipoxygenase (5-LO) and cyclooxygenase, effectively reduced inflammation in a murine model of acquired lymphedema (Jiang et al, 2018). The effect of HSFD on lipid mediators and the lymphedema immune microenvironment is unknown and will be the focus of future investigations.

SFAs are known to induce lipotoxicity, contributing to metabolic diseases such as insulin resistance and obesity (Nolan and Larter, 2009; Piccolis et al, 2019). In our study, SFAs, specifically SA and PA, induced significant apoptosis in LECs via mechanisms involving ROS generation and ER stress. Interestingly, vascular endothelial cells were less sensitive to these effects, suggesting a distinct susceptibility in LECs. In contrast to SFAs, the omega-6 PUFA LA failed to induce apoptosis. One key difference lies in cellular localization: LA binds to peroxisome proliferator-activated receptor gamma (PPARγ) in the nucleus, while SFAs predominantly localize in the cytosol within lipid droplets or the ER (Muralikumar et al, 2017). Additionally, SFAs contribute to β-oxidation in mitochondria, a process linked to increased mitochondrial ROS production following SA treatment (Adeva-Andany et al, 2019; Muralikumar et al, 2017), underscoring the central role of mitochondrial dysfunction in SFA-induced lipotoxicity.

Fatty acid binding proteins (FABPs) are essential lipid chaperones that regulate intracellular fatty acid trafficking and metabolism in many different cell types, including LECs. FABP4 binds both SFAs and PUFAs and has been implicated in primary lymphedema, with mutations near its nuclear localization signal suggesting a functional role in disease pathogenesis (Ayers et al, 2007; Ferrell et al, 2008). Under oxidative stress, FABP4 preferentially binds PA, potentially altering its function (Furuhashi, 2019). Our results demonstrate that FABP4 inhibition with BMS-309403 significantly reduced SA-induced apoptosis in LECs and ameliorated tail swelling in HSFD-fed mice. We also observed elevated FABP4 levels in plasma from both breast cancer-related and non-breast cancer lymphedema patients, suggesting FABP4 as a potential therapeutic target across diverse patient populations. FABP4 also contributes to cancer proliferation, and FABP4 inhibitors have been demonstrated to block tumor growth in

xenograft mice (Zeng et al, 2020). In our study, we found that plasma FABP4 levels in breast cancer-related lymphedema patients remained elevated in individuals who no longer had active cancer, similar to the high circulating FABP4 levels reported in studies evaluating breast cancer patients before initiating cancer treatments (Guaita-Esteruelas et al, 2017; Tsakogiannis et al, 2021). Surprisingly, the 15 breast cancer patients in our control cohort, who did not develop lymphedema within 5 years of monitoring, exhibited FABP4 levels comparable to those of healthy individuals. Given that FABP4 is typically elevated in breast cancer, this finding raises the possibility that circulating FABP4 could potentially serve as a specific marker for lymphedema after cancer remission. Furthermore, FABP4 inhibitors are in clinical development to treat breast cancer patients. While FABP4 inhibition appears beneficial for obese or diabetic patients, its impact on normal weight individuals remains unclear and could lead to metabolic and inflammatory imbalances. Further studies are needed to better understand these potential side effects. FABP4 plays a key role in fatty acid transport and metabolism in adipocytes and macrophages. Its inhibition could disrupt normal lipid regulation in many different cell types, leading to abnormal fatty acid accumulation, potentially affecting the liver and other metabolic tissues. Inhibiting FABP4 could increase circulating fatty acids, contributing to elevated triglycerides and atherogenic lipoproteins. Nevertheless, given its dual role in promoting cancer proliferation and contributing to lymphedema, FABP4 inhibition represents a compelling therapeutic strategy with the potential to address both cancer-related lymphedema and the metabolic alterations associated with cancer progression.

In conclusion, our study identifies SFAs as a potential driver of microenvironmental stress in lymphedema tissue, leading to ER stress, oxidative stress, and apoptosis. The prolonged tail swelling observed in HSFD-fed mice was prevented by dietary modifications and attenuated by FABP4 inhibition, suggesting these as viable therapeutic strategies. The PUFA/SFA ratio may serve as a potential biomarker for lymphedema risk, guiding dietary interventions and targeted therapies. Future studies will focus on further elucidating the molecular interactions between fatty acids, FABP4, and the lymphatic microenvironment to develop optimized treatments for lymphedema.

## Methods

**Reagents and tools table**

| Reagent/resource | Reference or source | Identifier or catalog number |
|---|---|---|
| **Experimental models** | | |
| C57BL6/J (*M. musculus*) | Charles River Laboratories | 027 |
| HDLECs (*H. sapiens*) | PromoCell | Cat# C-12217 |
| HUVECs (*H. sapiens*) | Thermo Fisher Scientific | Cat# C0035C |
| Rat Mesenteric Lymphatic Endothelial Cells (*R. norvegicus*) | | Prof. Pierre-Yves von der Weid, Calgary, Canada |
| HMEC-1 (*H. sapiens*) | ATCC | Cat# CRL-3243 |
| **Diets** | | |
| Chow diet | PicoLab Rodent Diet 20, LabDiet | 5053 |

| Reagent/resource | Reference or source | Identifier or catalog number |
|---|---|---|
| High fat diet | Research Diets Inc | D12492I |
| High saturated fat diet | Research Diets Inc | D12113001I |
| **Antibodies** | | |
| Rabbit anti-LYVE-1 | Thermo Fisher Scientific | Cat# PA1-16636 |
| Rat anti-LYVE-1 | Thermo Fisher Scientific | Cat# 14-0443-80 |
| Mouse anti-8-OHdG | Abcam | Cat# ab48508 |
| Rat anti-F4/80 | Thermo Fisher Scientific | Cat# MA1-91124 |
| Rat anti-CD4 | Thermo Fisher Scientific | Cat# 14-0041-82 |
| Rabbit anti-sXBP-1 | Cell Signaling Technology | Cat# 40435 |
| Mouse anti-CHOP | Cell Signaling Technology | Cat# 2895 |
| Mouse anti-FABP4 | Thermo Fisher Scientific | Cat# MA5-49201 |
| Rabbit anti-FABP5 | Thermo Fisher Scientific | Cat# PA5-92929 |
| Rabbit anti-Calreticulin | Thermo Fisher Scientific | Cat# MA5-51367 |
| Rabbit anti-Mcl-1 | Cell Signaling Technology | Cat# 94296 |
| Mouse anti-Bcl-2 | Cell Signaling Technology | Cat# 15071 |
| Rabbit anti-Caspase-3 | Cell Signaling Technology | Cat# 9662 |
| Rabbit anti-PARP1 | Cell Signaling Technology | Cat# 9542 |
| Anti-GAPDH | Thermo Fisher Scientific | Cat# AM4300 |
| Anti-α-Tubulin | Cell Signaling Technology | Cat# 2125 |
| Goat anti-mouse Alexa Fluor 488 | Thermo Fisher Scientific | Cat# A32723 |
| Goat anti-mouse Alexa Fluor 555 | Thermo Fisher Scientific | Cat# A32727 |
| Goat anti-mouse Alexa Fluor 647 | Thermo Fisher Scientific | Cat# A48289 |
| Goat anti-rabbit Alexa Fluor 488 | Thermo Fisher Scientific | Cat# A32731 |
| Goat anti-rabbit Alexa Fluor 647 | Thermo Fisher Scientific | Cat# A32733 |
| Goat anti-rat Alexa Fluor 555 | Thermo Fisher Scientific | Cat# A48263 |
| Goat anti-mouse Alexa Fluor 647 | Abcam | Cat# Ab150115 |
| HRP goat anti-mouse IgG | LiCor | 926-80010 |
| HRP goat anti-rabbit IgG | LiCor | 926-80011 |
| **Chemicals, enzymes and other reagents** | | |
| BMS-309403 | Tocris Bioscience | Cat# 5258/10 |
| BMS-309403 sodium | MedChemExpress | Cat# HY-*101903*A |

| Reagent/resource | Reference or source | Identifier or catalog number |
|---|---|---|
| Neutral buffered formalin, 10% | Fisher Scientific | Cat# 23-245684 |
| Ethylenediaminetetraacetic acid (EDTA) | Sigma-Aldrich | Cat# E5134 |
| Antigen Retrieval Solution | Thermo Fisher Scientific | Cat# 00-4955-58 |
| Triton X-100 | Fisher Scientific | Cat# AAA16046AE |
| Blocker™ BSA | Thermo Fisher Scientific | Cat# 37525 |
| Mouse-on-Mouse IgG Blocking Solution | Thermo Fisher Scientific | Cat# R37621 |
| VECTASHIELD® Antifade Mounting Medium | Vector Laboratories | Cat# H-1000-10 |
| Paraplast | Leica Biosystems | Cat# 39601006 |
| Methanol | Sigma-Aldrich | Cat# 34860-4L-R |
| 4′,6-diamidino-2-phenylindole (DAPI) | Sigma-Aldrich | Cat# D9542 |
| Gelatin | Sigma-Aldrich | Cat# G1890 |
| Endothelial Cell Growth Medium MV 2 Kit | PromoCell | Cat# C-22121 |
| MCDB-131 medium | Thermo Fisher Scientific | Cat# 12483-012 |
| Dulbecco's Modified Eagle's Medium (DMEM) | Sigma-Aldrich | Cat# D5030 |
| Human Epidermal Growth Factor | Sigma-Aldrich | Cat# E9644 |
| Hydrocortisone | Sigma-Aldrich | Cat# H0396 |
| L-glutamine | Gibco | Cat# 25030081 |
| Fetal bovine serum (FBS) | Gibco | Cat# 16000069 |
| Stearic acid | Sigma-Aldrich | Cat# 85679 |
| Palmitic acid | Fisher Scientific | Cat# AC129702500 |
| Linoleic acid | Fisher Scientific | Cat# AC215040250 |
| Fatty acid-free BSA | Sigma-Aldrich | Cat# A3803 |
| α-tocopherol | Sigma-Aldrich | Cat# 258024 |
| Hydrogen peroxide 30% | Fisher Scientific | Cat# H325-500 |
| Ethanol | Greenfield Global | Cat# P016EAAN |
| Dimethyl sulfoxide (DMSO) | Sigma-Aldrich | Cat# D2650 |
| Sodium Hydroxide Solution | Fisher Scientific | Cat# SS255-1 |
| Trypan blue | Sigma-Aldrich | Cat# T8154 |
| Annexin V | BD Canada | Cat# 556419 |
| 7-aminoactinomycin D (7-AAD) | BD Canada | Cat# 559925 |
| Crystal violet | Sigma-Aldrich | Cat# C6158 |
| Paraformaldehyde, 4% | Santa Cruz Biotechnology | Cat# sc-281692 |
| Oil Red O | Fisher Scientific | Cat# AAA1298922 |
| Isopropanol | Sigma-Aldrich | Cat# 439207 |
| Modified Harris Hematoxylin | Fisher Scientific | Cat# 22-050-206 |
| Eosin Y solution | Sigma-Aldrich | Cat# 318906 |

| Reagent/resource | Reference or source | Identifier or catalog number |
|---|---|---|
| Dihydroethidium (DHE) | Thermo Fisher Scientific | Cat# D11347 |
| MitoSox Red | Thermo Fisher Scientific | Cat# M36008 |
| Protease and phosphatase inhibitor cocktail | Thermo Fisher Scientific | Cat# 78440 |
| Clarity Western ECL substrate | Bio-Rad | Cat# 170-5060 |
| Hydrogen peroxide | Fisher Scientific | Cat# H325-500 |
| Type I human collagen | Sigma-Aldrich | Cat# CLS354265 |
| NP-40 lysis buffer | Thermo Fisher Scientific | Cat# J60766.AP |
| **Kits** | | |
| Click-iT™ Plus TUNEL Apoptosis Assay Kit | Thermo Fisher Scientific | Cat# C10618 |
| Human FABP4/A-FABP ELISA Kit | Thermo Fisher Scientific | Cat# EH177RB |
| Mouse FABP4/A-FABP ELISA Kit | Novus Biologicals | Cat# NBP2-82410 |
| Pierce BCA Protein Assay Kit | Thermo Fisher Scientific | Cat# 23225 |
| **siRNA transfection** | | |
| FABP4-targeting siRNA | Invitrogen | Cat# s4964 |
| Silencer Select Negative Control | Invitrogen | Cat# 4390843 |
| Lipofectamine RNAiMAX | Invitrogen | Cat# 13778075 |
| Opti-MEM | Gibco | Cat# 31985062 |
| **Other** | | |
| 7890A Gas Chromatograph | Agilent Technologies | G3440A |
| CP-Sil 88 GC Column | Agilent Technologies | CP7489 |
| Traceable™ Digital Carbon Fiber Caliper | Fisher Scientific | S90187A |
| ZEISS Axioplan 2 light microscope | ZEISS Germany | B-290TB |
| ZEISS Axio Imager 2 fluorescent microscope | ZEISS Germany | |
| Attune NxT flow cytometer | Thermo Fisher Scientific | A28997 |
| Trans-Blot Turbo | Bio-Rad | 1704150 |
| UVP ChemStudio gel imaging system | Analytikjena | 76209-532 |
| AutoScratch Wound Making Tool | Agilent Technologies | 750011 |
| BioSpa 8 Automated Incubator | Agilent Technologies | 1-120-552 |
| Cytation 10 Cell Imaging Reader | Agilent Technologies | BTC10MPWC |
| **Software** | | |
| GraphPad Prism | https://www.graphpad.com | Version 10.2.3 |

| Reagent/resource | Reference or source | Identifier or catalog number |
|---|---|---|
| Fiji | https://imagej.net/software/fiji/ | Version 1.54 f |
| FlowJo | Tree Star Inc. | Version 10.10 |
| BioSpa OnDemand | Agilent Technologies | Version 1.04 |

## Study approval

Studies utilizing human samples were approved by the Ethics Committee of the University of Alberta (Pro00115131, 2024_AB241128). Written, informed consent was obtained from all study subjects prior to participation. All human research procedures were conducted in accordance with the principles of the WMA Declaration of Helsinki and the Department of Health and Human Services Belmont Report. Patients were recruited through rehabilitation clinics within Alberta Health Services and the Alberta Cancer Research Biobank. Breast cancer control patients were followed for 5 years post cancer treatment and did not develop lymphedema during the monitoring period. For animal studies, all experiments and procedures were approved by the Animal Care and Use Committee of the University of Alberta (AUP4047).

## Plasma fatty acid profile

Blood samples from non-fasted participants (including both non-lymphedema and lymphedema groups) were centrifuged at $1500 \times g$ for 15 min at 4 °C, and plasma sample aliquots were stored at −80 °C for subsequent analysis. A modified Folch extraction method (Field et al, 1988; Folch et al, 1957) was employed to isolate lipids from the plasma samples. Briefly, total lipids were extracted and the lipid classes separated by thin-layer chromatography. Internal standards were added for quantification and identification of individual lipid peaks (Abdelmagid et al, 2015). Lipid fractions were methylated (boron trifluoride and hexane at 100 °C) and fatty acids were separated and quantified by automated GLC 7890A (Agilent Technologies) on a CP-Sil 88 column (100 m × 0.25 mm; Agilent) (Cruz-Hernandez et al, 2013). Fatty acids were expressed as the relative percentage of total fatty acid content.

## Mice

Female C57BL/6J mice (6 weeks old) were obtained from Charles River Laboratories and housed at the Health Sciences Laboratory Animal Services (HSLAS) facility at the University of Alberta. Mice were maintained in a temperature- and humidity-controlled environment with a 12/12-h reverse light-dark cycle. After a 2-week acclimation period on a standard CD (5053, PicoLab Rodent Diet 20, LabDiet), mice were randomly assigned to one of three dietary groups: CD, HFD (D12492, Research Diets Inc) or HSFD (D12113001, Research Diets Inc). Both the HFD and HSFD provided 60% of their energy from fat. However, the primary fat source of the HFD consisted of lard, while the HSFD was predominantly composed of cocoa butter. The detailed composition of each diet is provided in Appendix Table S2.

Mice were fed their respective diets for 4 weeks before the surgical intervention and remained on these diets for an additional 4 weeks post-surgery. To determine the effects of pharmacological inhibition of FABP4, either the FABP4 inhibitor BMS-309403 (15 mg·kg$^{-1}$·day$^{-1}$; HY-101903A, MedChemExpress, NJ, USA) or vehicle (phosphate-buffered saline, PBS) was administered chronically by daily oral gavage for 4 weeks starting on the day of the lymphatic surgery.

## Tail lymphedema model

Acquired lymphedema was surgically induced in mice by ablating the lymphatic trunks in the tail. Mice were anesthetized with 2–3% isoflurane and maintained on a heating pad to keep their body temperature at 36.5–37.5 °C. A 3-mm circumferential incision was made 2 cm from the base of the tail, and the lymphatic trunks were ablated using microsurgical scissors under a surgical microscope. Only the skin was incised in the sham surgery group, leaving the primary collecting lymphatics intact. Mice showing signs of tail necrosis or infection post-surgery were excluded. Tail circumference was measured every other day using a digital caliper at 5-mm intervals starting from the surgical site. The tail volume was calculated using the truncated cone equation. Mice were euthanized 28 days after surgery, and tails, along with cardiac blood, were collected for further analysis.

## Hematoxylin and eosin staining

To evaluate tissue morphology in the context of lymphedema, tail tissues were fixed overnight in 10% neutral buffered formalin (Fisher Scientific, Ottawa, ON, Canada), decalcified using 10% ethylenediaminetetraacetic acid (EDTA; Sigma-Aldrich, St. Louis, MO, USA), and embedded in paraffin. H&E staining was performed on 5-μm sections according to standard histological protocols. Stained slides were examined using a ZEISS Axioplan 2 (ZEISS, Oberkochen, Germany) light microscope.

## Immunofluorescence staining

Paraffin-embedded tail sections (5 μm) were rehydrated and subjected to heat-mediated antigen retrieval using IHC antigen retrieval solution (Thermo Fisher Scientific). Sections were incubated overnight at 4 °C with primary antibodies against the following markers: LYVE-1 (PA1-16636, Thermo Fisher Scientific; 1:400), 8-OHdG (ab48508, Abcam, Cambridge, UK; 1:100), sXBP-1 (40435, Cell Signaling Technology, Danvers, MA, USA; 1:200), CHOP (2895, Cell Signaling Technology; 1:200), F4/80 (MA1-91124, Thermo Fisher Scientific; 1:200), CD4 (14-0041-82, Thermo Fisher Scientific), and/or FABP4 (MA5-49201, Thermo Fisher Scientific; 1:200). Secondary antibodies were labeled with the fluorochromes Alexa Fluor 488, Alexa Fluor 555, or Alexa Fluor 647 (Thermo Fisher Scientific; 1:500). Nuclei were stained with 4′,6-diamidino-2-phenylindole (DAPI) (Sigma-Aldrich). Immunofluorescence (IF) slides were imaged using a ZEISS Axio Imager 2 (ZEISS, Oberkochen, Germany) fluorescence microscope. Fluorescence image acquisition was performed with fixed acquisition settings across all experimental groups. Exposure time and laser power for each fluorophore channel were initially calibrated using samples with the highest expected fluorescence intensity. These standardized settings were applied to all samples in each experiment to ensure analytical consistency and to avoid post-acquisition bias.

Image analysis was performed using Fiji software version 1.54f (National Institutes of Health, Bethesda, MD, USA). A minimum of five high-powered fields per animal were analyzed. For fluorescence analysis, the percentage of positive area was quantified relative to the total DAPI-stained nuclear area. LYVE-1$^+$ lymphatic vessels were first identified morphologically and outlined manually using the polygon selection tool in Fiji to generate vessel-specific regions of interest (ROIs). A concentric perilymphatic area extending 100 μm from each vessel was created using the "Enlarge" function. Within this defined zone, binary masks were generated for each marker channel after consistent thresholding. The "Measure" function in Fiji was used to calculate the total positive area, and values were expressed as a percentage of the total nuclear (DAPI) area in the corresponding ROI. This approach allowed normalization for cell density and tissue section variability. To evaluate marker expression within lymphatic endothelial cells (LECs), colocalization analysis was performed within the LYVE-1$^+$ vessel ROIs. Colocalization of each marker with LYVE-1 was assessed using the "Coloc 2" plugin, which computes Pearson's correlation coefficient. Channels were aligned and background-subtracted before analysis. Nuclear colocalization was further validated by confirming overlap with DAPI staining. Given that LYVE-1 can also label subsets of macrophages, additional morphological features such as vessel shape and lumen presence were used to distinguish true lymphatic structures. In select experiments, F4/80 was used as a secondary macrophage marker to validate lymphatic endothelial identity.

Adipocyte size and number were quantified in sections stained for FABP4, a cytoplasmic marker of mature adipocytes. Adipocytes were identified based on FABP4$^+$ staining, large cytoplasmic area, and the presence of a peripherally displaced DAPI$^+$ nucleus. Mean adipocyte area was calculated per field. Adipocyte number was expressed as the total number of FABP4$^+$ adipocytes per high-powered field. At least five fields per animal were analyzed.

Cells were fixed with 4% paraformaldehyde (sc-281692, Santa Cruz Biotechnology) in PBS for 15 min at room temperature. Following fixation, cells were washed and incubated with primary antibodies against FABP4 (1:200) and calreticulin (MA5-51367, Thermo Fisher Scientific; 1:200). After overnight incubation with primary antibodies, appropriate species-specific secondary antibodies conjugated to Alexa Fluor 488 or Alexa Fluor 647 (Thermo Fisher Scientific; 1:500) were applied. Nuclei were counterstained with DAPI. Fluorescent images were captured using a ZEISS Axio Imager 2 microscope. For each experimental condition, at least four randomly selected, non-overlapping fields were acquired per replicate, with three independent biological replicates analyzed. Colocalization between calreticulin and FABP4 was assessed by calculating Pearson correlation coefficients using the Coloc2 plugin in Fiji. Before analysis, fluorescence channels were aligned, and background signals were subtracted to minimize artifacts and ensure robust quantification.

## TUNEL assay

To determine the amount of cell death in lymphedema tissue, the TUNEL method was performed to label apoptotic DNA fragmentation in the tail sections. The TUNEL assay was carried out with the

Click-iT™ Plus TUNEL Apoptosis Assay Kit (C10618, Thermo Fisher Scientific) combined with Anti-LYVE-1 (1:400), according to the manufacturer's instructions. The images of TUNEL-positive cells were captured using a ZEISS Axio Imager 2 fluorescent microscope and subsequently quantified using Fiji software.

## Cell culture and treatments

Three different primary endothelial cell lines were utilized in this study: Human Dermal Lymphatic Endothelial Cells (HDLECs; C-12217, PromoCell, Heidelberg, Germany), derived from juvenile foreskin (donor sex not specified); Human Umbilical Vein Endothelial Cells (HUVECs; C0035C, Thermo Fisher Scientific), isolated from a single male newborn donor (≤14 days old); and Rat Mesenteric Lymphatic Endothelial Cells (RMLECs), generously provided by Dr. Pierre-Yves von der Weid (University of Calgary, Calgary, Canada). All primary cell lines were cultured on tissue culture plates coated with 0.2% gelatin (Sigma-Aldrich) and maintained in Endothelial Cell Growth Medium-2 (ECGM-2; C-22121, PromoCell) supplemented with the manufacturer's recommended additives, including 5% fetal bovine serum (FBS). Cells were incubated at 37 °C in a humidified atmosphere containing 5% $CO_2$ and used for experiments between passages 5 and 8. Additionally, the immortalized Human Dermal Microvascular Endothelial Cells (HDMECs; CRL-3243, ATCC, Manassas, VA, USA), derived from the foreskin of a male infant (<1 month old), were cultured in MCDB-131 medium supplemented with 5% FBS (12483-012, Thermo Fisher Scientific), 10 ng/mL human epidermal growth factor (hEGF; E9644, Sigma-Aldrich), 1 µg/mL hydrocortisone (H0396, Sigma-Aldrich), and 10 mM L-glutamine (25030081, Gibco). HDMECs were also seeded on 0.2% gelatin-coated plates and maintained at 37 °C with 5% $CO_2$ in a humidified incubator. Prior to treatment, all cell lines were incubated overnight in medium containing 1% FBS. Drug treatments were performed when cultures reached approximately 70–80% confluency, corresponding to a subconfluent state. Representative phase-contrast images of cell morphology at the time of treatment are provided in Appendix Fig. S23A. All in vitro experiments were performed in at least three independent replicates.

Stock solutions of the saturated fatty acids stearic acid (SA; 85679, Sigma-Aldrich) and palmitic acid (PA; AC129702500, Fisher Scientific, Ottawa, ON, Canada) were prepared by conjugation to fatty acid-free bovine serum albumin (BSA) (A3803, Sigma-Aldrich) to enhance solubility and cellular bioavailability in aqueous media. Fatty acids were first dissolved in a minimal volume of 0.1 M NaOH, heated to 70 °C until fully solubilized, and then added to prewarmed (55 °C) 10% BSA solution. The mixture was incubated at 55 °C for 30 min with constant stirring to facilitate complete conjugation. Cells were treated with stearic acid at final concentrations ranging from 1 to 100 µM, and with palmitic acid at concentrations ranging from 1 to 500 µM, for 24 or 48 h. The polyunsaturated fatty acid linoleic acid (LA; AC215040250, Fisher Scientific) was handled and stored under a nitrogen atmosphere in amber glass vials to prevent oxidation. LA was added dropwise to prewarmed BSA solution with gentle stirring under nitrogen until fully incorporated. All fatty acid–BSA complexes were prepared at a final molar ratio of 4:1, a condition optimized to reflect physiological relevance and ensure consistent delivery across experimental conditions. Standardized conjugation protocols were used throughout the study to ensure reproducibility across batches and fatty acid species.

Stock solutions of the antioxidant α-tocopherol (258024, Sigma-Aldrich) were prepared in ethanol, while the FABP4 inhibitor BMS-309403 (5258, Tocris Bioscience, Bristol, UK) was dissolved in DMSO (D2650, Sigma-Aldrich). Hydrogen peroxide ($H_2O_2$) (H325-500, Fisher Scientific) was diluted in sterile water and used as a pre-treatment to mimic oxidative stress conditions. Working dilutions of all compounds were freshly prepared in cell culture media immediately prior to use. Pre-treatments with α-tocopherol (200 µg/mL) and $H_2O_2$ (300 µM) were administered 1 h before saturated fatty acid exposure, while LA (50 µM) and BMS-309403 (5 µM) were added 30 min prior to treatment.

## Cell viability assays

Cell viability was assessed using multiple assays to evaluate different aspects of cell death and survival. The trypan blue exclusion assay was used to measure cell viability by staining cells with 0.04% trypan blue (T8154, Sigma-Aldrich). After treatments, cells and cell culture media containing dead floating cells were collected, resuspended in PBS, and stained with trypan blue at a 1:1 ratio. Viable cells remained unstained, while non-viable cells were stained blue. Live and dead cells were counted using a hemocytometer, and total cell death was calculated as total cell death (%) = dead cells/(live + dead cells) × 100. Representative images of stained cells acquired during the assay are shown in Appendix Fig. S23B. The annexin V/7-AAD assay quantified apoptosis and cell death by staining cells with annexin V-FITC and 7-AAD (BD Biosciences, San Jose, CA, USA). Apoptotic cells were identified by annexin V positivity, and data were acquired using the Attune NxT flow cytometer (Life Technologies, Thermo Fisher Scientific). A total of 10,000 events were analyzed using FlowJo software (version 10.10, Tree Star Inc.). Lastly, the clonogenic assay was performed to evaluate long-term cell survival. Cells were seeded at low density, cultured for 7–14 days to allow colony formation, then fixed with 4% paraformaldehyde, stained with 0.5% crystal violet (C6158, Sigma-Aldrich), and manually counted. Colonies were defined as clusters containing ≥50 cells.

## Oil Red O staining

Lipid accumulation was assessed using Oil Red O (ORO) staining. For cultured cells, after treatment, cells were washed with PBS, fixed in 4% paraformaldehyde for 15 min at room temperature, air-dried, and stained with freshly filtered ORO solution (AAA1298922, Fisher Scientific) for 30 min, following the manufacturer's protocol. Excess stain was removed with 60% isopropanol, and lipid droplets were visualized using a bright-field microscope. For mouse tail tissue, 10-µm cryosections were air-dried at room temperature, fixed in 10% neutral buffered formalin for 10 min, and stained with freshly filtered ORO solution for 30 min. Slides were washed in 60% isopropanol (439207, Sigma-Aldrich), rinsed with distilled water, and counterstained with hematoxylin. All slides were mounted using aqueous medium and imaged under a light microscope using standardized settings to enable qualitative and semi-quantitative comparisons.

## ROS detection

After treatment, cells were harvested with the culture media, resuspended in PBS, and stained with 3.2 μM dihydroethidium (DHE, D11347, Thermo Fisher Scientific) for 30 min at 37 °C to assess total ROS levels. For detection of mitochondrial superoxide, cells were stained with 1 μM MitoSox Red (M36008, Thermo Fisher Scientific) for 30 min at 37 °C. ROS levels were quantified using an Attune NxT flow cytometer, and data were analyzed using FlowJo software (version 10.10). A total of 10,000 events were collected per sample, and ROS levels were measured based on fluorescence intensity.

## Western blotting

To assess protein expression, protein lysates were prepared using NP40 lysis buffer supplemented with protease and phosphatase inhibitors (78440, Thermo Fisher Scientific). Protein concentrations were determined using the Pierce BCA Protein Assay Kit (23225, Thermo Fisher Scientific), and measurements were taken with a FLUOstar Omega multi-plate reader (BMG Labtech, Ortenberg, Germany). Equal amounts of 20–30 μg of protein were loaded onto a Tris-glycine SDS-PAGE gel and separated by electrophoresis. Proteins were then transferred to nitrocellulose membranes using the Trans-Blot Turbo (Bio-Rad, Hercules, USA). Membranes were incubated overnight at 4 °C with primary antibodies diluted in 0.1% TBS-T, followed by a 1-h incubation with HRP-conjugated secondary antibodies. Protein bands were detected using Clarity Western ECL substrate (170-5060, Bio-Rad) and imaged with the UVP ChemStudio gel imaging system (Analytikjena, Upland, CA, USA). Quantification of protein expression was performed using Fiji software version 1.54 f (National Institutes of Health, Bethesda, MD, USA). The following primary antibodies were used: Anti-sXBP-1 (40435, Cell Signaling Technology; 1:1000), Anti-CHOP (2895, Cell Signaling Technology; 1:1000), Anti-Mcl-1 (94296, Cell Signaling Technology, Danvers, MA, USA; 1:1000), Anti-Bcl-2 (15071, Cell Signaling Technology; 1:1000), Anti-Caspase 3 (9662, Cell Signaling Technology; 1:500), Anti-PARP1 (9542, Cell Signaling Technology; 1:1000), Anti-FABP4 (MA5-49201, Thermo Fisher Scientific; 1:1000), Anti-FABP5 (PA5-92929, Thermo Fisher Scientific; 1:1000), Anti-GAPDH (AM4300, Thermo Fisher Scientific; 1:5000), and Anti-alpha-Tubulin (2125, Cell Signaling Technology; 1:5000). Secondary antibodies of HRP goat anti-mouse IgG (926-80010; 1:10,000) and HRP goat anti-rabbit IgG (926-80011; 1:10,000) were purchased from LI-COR Biotechnology (Lincoln, NE, USA).

## In vitro scratch assay

Endothelial cell migration was evaluated using a high-throughput scratch wound assay in 96-well plates. HDLECs and HDMECs were seeded to confluency and uniformly scratched using the AutoScratch Wound Making Tool (Agilent Technologies). Detached cells were removed by gentle washing with PBS, and fresh culture medium containing either vehicle or 10 μM SA was added. Plates were transferred to the BioSpa 8 Automated Incubator (Agilent Technologies) integrated with the Cytation 10 Cell Imaging Reader (Agilent Technologies), and high-contrast bright-field images were acquired every 4 h over 48 h using BioSpa OnDemand Software v1.04. Wound area was quantified using Fiji software with the Wound Healing Size

Tool plugin, which enabled automated detection and measurement of the wound area across time points.

## Lymphatic ring assay

Lymphatic sprouting was assessed using an ex vivo ring assay adapted from a previously described protocol (Bruyere et al, 2008). Briefly, mouse lymphatic thoracic trunks were carefully dissected, cleaned of surrounding adipose tissue, and cut into ~1 mm segments under a stereomicroscope. Type I collagen (1 mg/mL) was prepared in DMEM, and the pH was adjusted to 7.4. The collagen solution was dispensed into 96-well plates and allowed to polymerize at 37 °C. Lymphatic thoracic duct segments were embedded in the polymerized collagen and overlaid with MCDB-131 medium supplemented with 2.5% FBS. Cultures were maintained at 37 °C in a humidified 5% $CO_2$ incubator for 48 h before treatment with either vehicle or 10 μM SA. Lymphatic sprouting was monitored by phase-contrast microscopy between days 7 and 14.

## FABP4 ELISA

Plasma levels of FABP4 were quantified using species-specific ELISA kits following the manufacturer's protocols. The FABP4/A-FABP Human ELISA Kit (EH177RB, Thermo Fisher Scientific) was used for human samples, while the Mouse FABP4/A-FABP ELISA Kit (NBP2-82410, Novus Biologicals) was used for mouse samples. Both assays utilize a quantitative sandwich enzyme immunoassay format to ensure specific and sensitive detection of FABP4.

## siRNA-mediated FABP4 knockdown

FABP4 knockdown in HDLECs was performed using siRNA transfection. HDLECs were seeded onto 0.2% gelatin-coated plates in ECMV2 medium supplemented with 5% FBS and allowed to adhere overnight. On the following day, cells were transfected with 10 nM FABP4-targeting siRNA (Silencer Select Pre-Designed siRNA, s4964, Invitrogen) or negative control siRNA (Silencer Select Negative Control, 4390843, Invitrogen) using Lipofectamine RNAiMAX (Invitrogen, 13778075) in Opti-MEM (Gibco, 31985062), following the manufacturer's protocol. Briefly, siRNA and transfection reagent were pre-incubated in Opti-MEM for 15–20 min at room temperature before being added to cells cultured in serum-free ECMV2 medium. Cells were incubated with the transfection mix for 18 h. Following transfection, cells were washed and cultured in ECMV2 medium supplemented with 5% FBS for 24 h. The medium was then replaced with 1% FBS for an additional overnight incubation. Fatty acid treatment was carried out the next day by exposing cells to 10 μM SA or vehicle control for 16 h. Cells were harvested 96 h post-transfection, at which point FABP4 protein levels were significantly reduced, and processed for viability assays and western blot analysis.

## Statistics analysis

GraphPad Prism version 10.2.3. was used for statistical analysis. Pearson correlation analysis was used to determine the correlations between human plasma analytes and the immunostaining colocalizations. The significance was determined by unpaired $t$ tests. For the mouse data, differences between groups were identified using two-way ANOVA followed by Šídák's post hoc test. For the in vitro

## The paper explained

### Problem

Secondary lymphedema is a chronic and debilitating condition that impairs the lymphatic system, causing swelling, pain, recurrent infections, and reduced quality of life. While commonly triggered by cancer treatments, surgery, infections, or obesity, the role of dietary components, particularly saturated fatty acids (SFAs), in its progression remains poorly understood. Lipotoxicity, a detrimental consequence of excess SFAs, may contribute to lymphatic dysfunction, yet its underlying mechanisms and therapeutic interventions are not well-defined.

### Results

This study reveals that patients with secondary lymphedema have a reduced polyunsaturated fatty acid (PUFA) to SFA ratio, independent of body mass index. In lymphatic endothelial cells, stearic acid, a common SFA, triggers cell death, oxidative stress, and endoplasmic reticulum stress. Using a mouse model, a high SFA diet was shown to exacerbate lymphedema-related swelling and tissue damage after surgery, even in the absence of obesity. Importantly, transitioning to a standard diet post-surgery alleviated these effects. Elevated levels of fatty acid-binding protein 4 (FABP4) were also detected in patients with lymphedema. Inhibiting FABP4 reduced stearic acid-induced damage in cells and mitigated lymphedema-related swelling and tissue injury in mice.

### Impact

These findings highlight the detrimental role of SFAs in lymphedema progression through lipotoxicity, providing critical insights into its pathophysiology. This research suggests that dietary interventions to reduce SFAs and therapeutic targeting of FABP4 may offer new strategies to manage or prevent lymphedema, improving outcomes and quality of life for affected patients.

data, statistical significance was determined using one-way ANOVA followed Tukey's post hoc test or an unpaired $t$ test. Data are presented as mean ± standard error of the mean (SEM) and $P$ values lower than 0.05 were considered statistically significant. Exact $P$ values for the statistical comparisons are shown in Appendix Table S3.

## Graphics

The synopsis graphic and Appendix Fig. S5A were created using BioRender.com.

## Data availability

This study includes no data deposited in external repositories.

The source data of this paper are collected in the following database record: biostudies:S-SCDT-10_1038-S44321-025-00286-4.

## Peer review information

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

## Acknowledgements

This research was supported by the University Hospital Foundation research grant (RES0056514). We are profoundly grateful to Dianne and Irving Kipnes for their extraordinary generosity and visionary leadership in supporting the University Hospital Foundation. Dianne Kipnes, who sadly passed away recently, dedicated her life to advancing learning, higher education, and innovative initiatives in healthcare and beyond. Her contributions will leave a legacy, and we are deeply honored to recognize her role in supporting this work and dedicate this paper in her memory. We also extend our gratitude to Dr. Pierre-Yves von der Weid (University of Calgary) for providing the RMLECs and his valuable scientific advice. Additionally, we acknowledge Dr. David Eisenstat for his critical review of the manuscript. Technical support from the University of Alberta's Flow Cytometry and Imaging Core Facilities is sincerely appreciated, as is the contribution of human plasma samples by the Alberta Cancer Research Biobank.

## Author contributions

**Karina P Gomes**: Data curation; Formal analysis; Methodology; Writing—original draft. **Jacob Korodimas**: Data curation; Methodology. **Emily Liu**: Data curation; Methodology. **Nirav Patel**: Data curation; Methodology. **Xiaoyan Yang**: Data curation; Methodology. **Susan Goruk**: Data curation; Methodology. **Jaqueline Munhoz**: Data curation; Methodology. **Catherine J Field**: Conceptualization; Supervision; Writing—review and editing. **Spencer B Gibson**: Conceptualization; Formal analysis; Supervision; Funding acquisition; Writing—review and editing.

Source data underlying figure panels in this paper may have individual authorship assigned. Where available, figure panel/source data authorship is listed in the following database record: biostudies:S-SCDT-10_1038-S44321-025-00286-4.

## Disclosure and competing interests statement

The authors declare no competing interests.

# Expanded View Figures

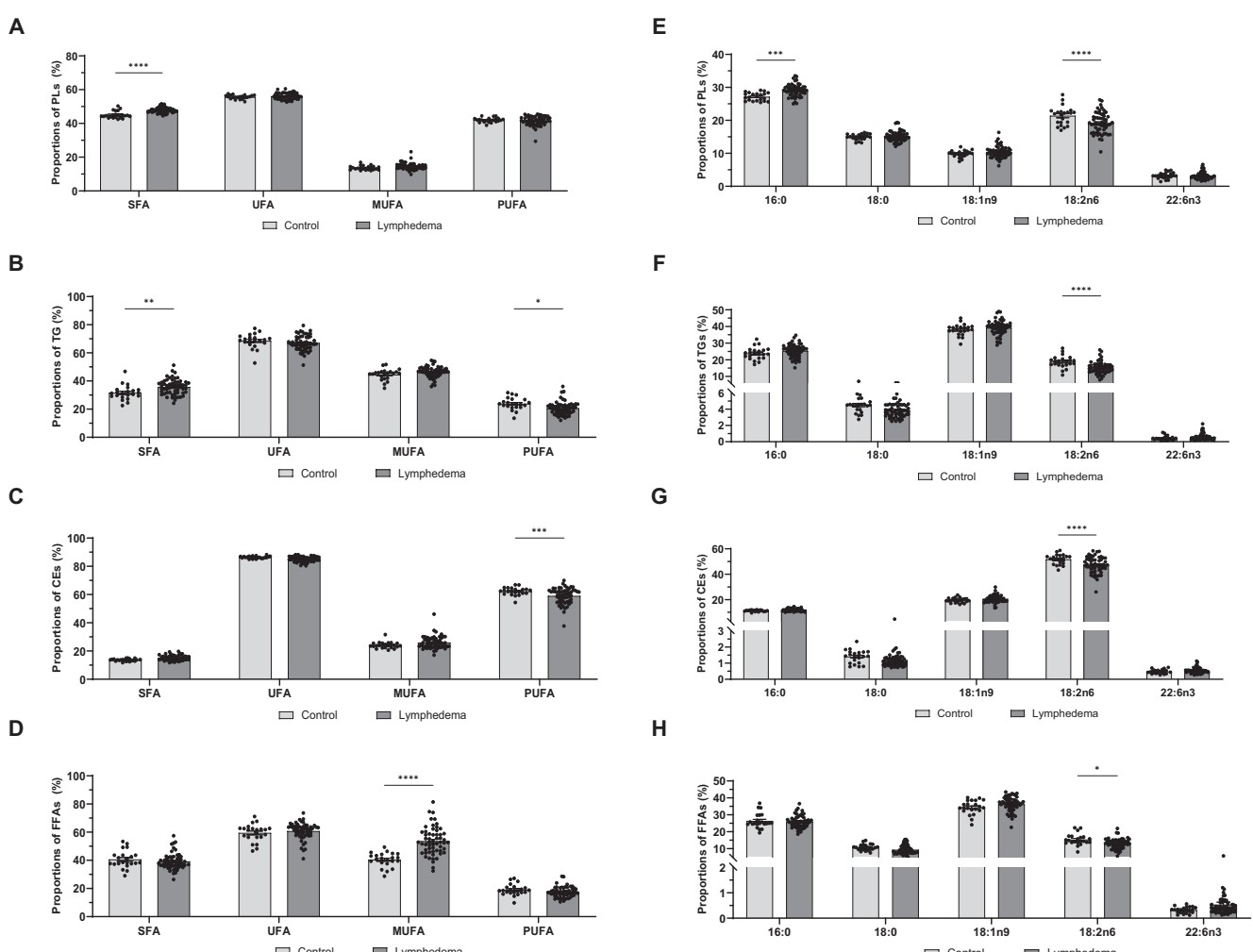

**Figure EV1. Fatty acid composition in plasma lipid fractions of patients with lymphedema.**

Plasma samples from patients with lymphedema ($n = 57$) and non-lymphedema controls ($n = 22$) were analyzed to assess the distribution of saturated (SFA), unsaturated (UFA), monounsaturated (MUFA), and polyunsaturated fatty acids (PUFA) across major lipid classes: (A) phospholipids (PLs), (B) triglycerides (TGs), (C) cholesterol esters (CEs), and (D) free fatty acids (FFAs). (E–H) Relative abundance of key fatty acid species—palmitic acid (16:0), stearic acid (18:0), oleic acid (18:1n9), linoleic acid (18:2n6), and docosahexaenoic acid (22:6n3)—within each lipid fraction: (E) PLs, (F) TGs, (G) CEs, and (H) FFAs. Data are presented as mean ± SEM. Statistical analysis: two-way ANOVA with Šídák's post hoc test. Significance: *$P < 0.05$, **$P < 0.01$, ***$P < 0.001$, ****$P < 0.0001$. Exact $P$ values for the statistical comparisons are shown in Appendix Table S3. Source data are available online for this figure.

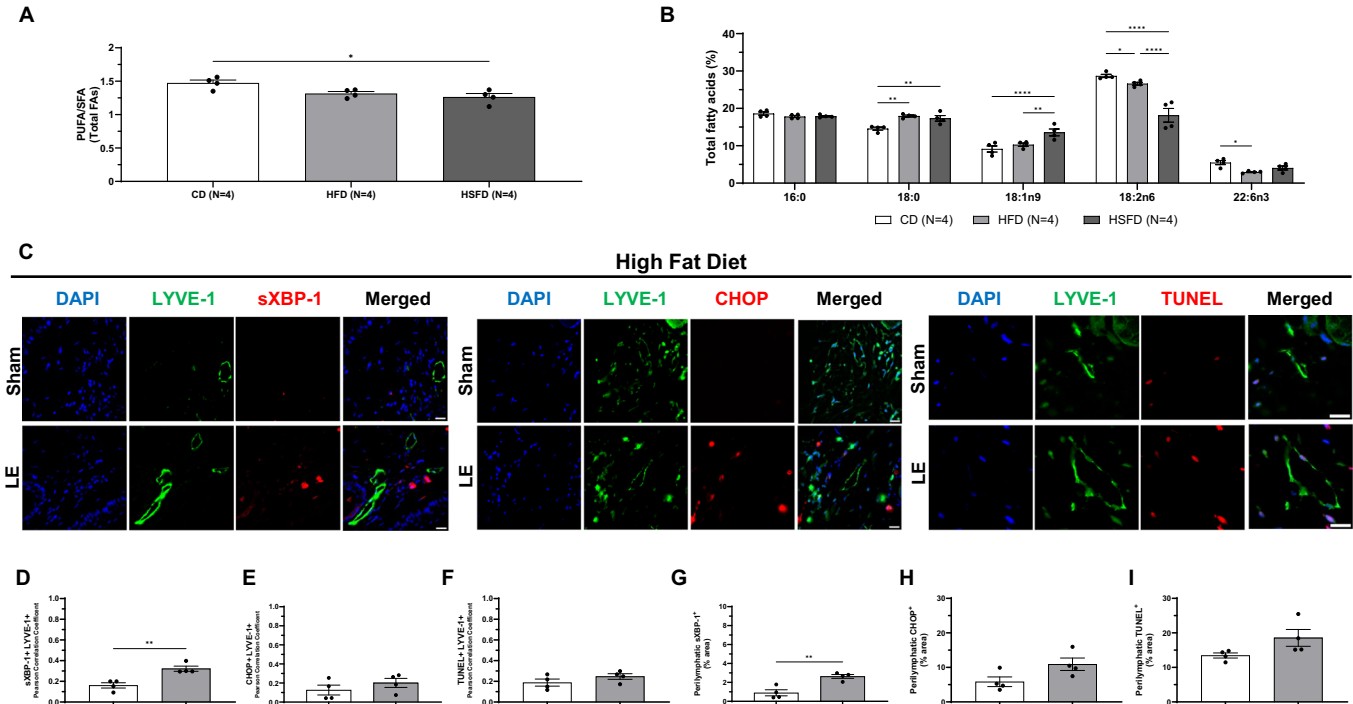

**Figure EV2. PUFA/SFA ratio and fatty acid composition in plasma total fatty acids of mice fed different diets.**

(A) PUFA/SFA ratio in plasma total fatty acids from mice fed a chow diet (CD), high-fat diet (HFD), or high saturated fat diet (HSFD) ($n = 4$ mice). (B) Proportions of individual fatty acids: palmitic acid (16:0), stearic acid (18:0), oleic acid (18:1n9), linoleic acid (18:2n6), and docosahexaenoic acid (22:6n3) ($n = 4$ mice). (C) Representative images of tail tissue from sham and lymphedema (LE) mice fed a high-fat diet, stained for LYVE-1 (green), sXBP-1, CHOP, or TUNEL (red), and DAPI (blue). (D–F) Quantification of sXBP-1, CHOP, and TUNEL signal colocalized with LYVE-1 positive cells ($n = 4$ mice). (G–I) Quantification of perilymphatic signal for sXBP-1, CHOP, and TUNEL, showing elevated ER stress and apoptosis in lymphedema tissue ($n = 4$ mice). Data are presented as mean ± SEM. Statistical analysis: one-way ANOVA with Tukey's post hoc test for (A); two-way ANOVA with Šídák's post hoc test for (B); two-tailed unpaired $t$ test for (D–I). Significance: *$P < 0.05$, **$P < 0.01$, ***$P < 0.001$, ****$P < 0.0001$. Exact $P$ values for the statistical comparisons are shown in Appendix Table S3. PUFA polyunsaturated fatty acid, SFA saturated fatty acid, LYVE-1 lymphatic vessel endothelial hyaluronan receptor-1, sXBP-1 spliced X-box binding protein 1, CHOP C/EBP homologous protein, TUNEL terminal deoxynucleotidyl transferase dUTP nick end labeling, DAPI 4′,6-diamidino-2-phenylindole. Scale bars: 50 μm. Source data are available online for this figure.

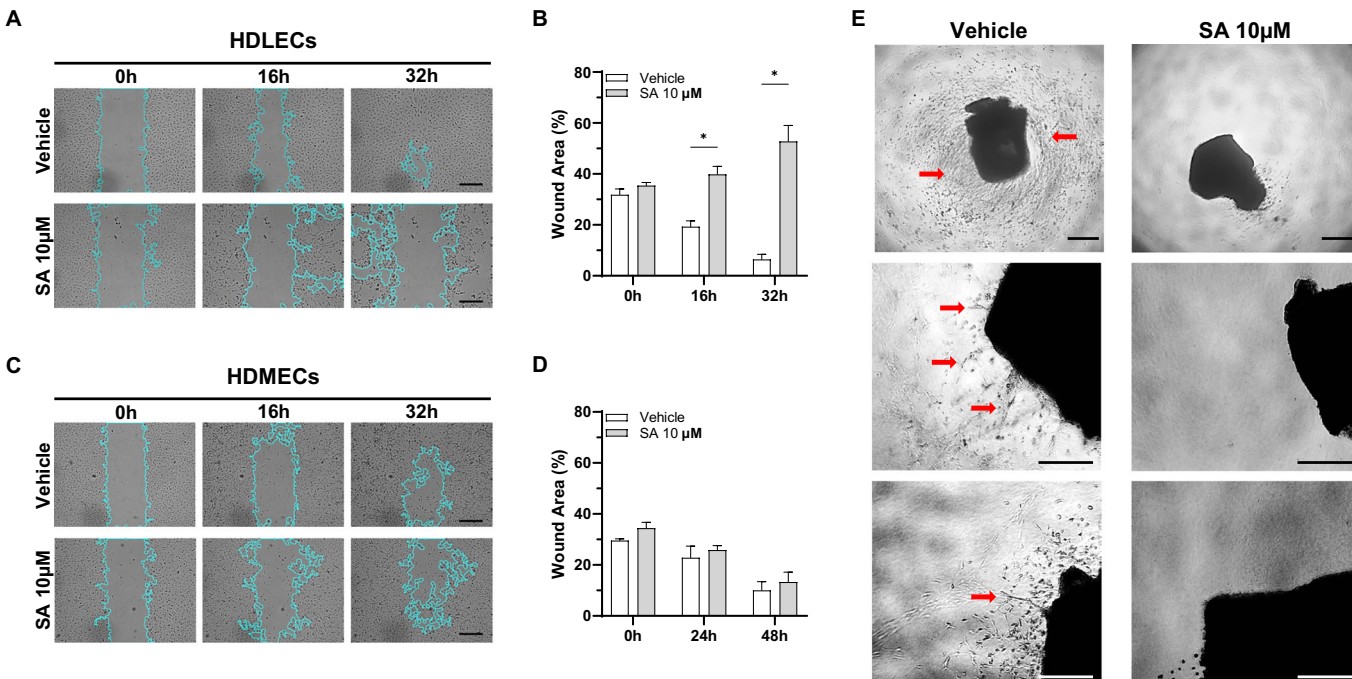

**Figure EV3. Stearic acid impairs lymphatic endothelial migration and vessel sprouting.**

(A–D) Scratch wound assay in human dermal lymphatic endothelial cells (HDLECs) and human dermal microvascular endothelial cells (HDMECs) treated with vehicle or stearic acid (SA, 10 µM). (A) Representative bright-field images of HDLECs at 0, 16, and 32 h post-scratch with wound edges outlined in cyan. Scale bars: 200 µm. (B) Quantification of wound area over time showing significantly impaired wound closure in SA-treated HDLECs (n = 4 independent experiments). (C) Representative bright-field images of HDMECs at 0, 24, and 48 h post-scratch with wound edges outlined in cyan. Scale bars: 200 µm. (D) Wound area quantification shows minimal effect of SA on HDMEC migration (n = 4 independent experiments). (E) Lymphatic ring assay using mouse thoracic duct segments embedded in collagen. Vehicle-treated segments show robust endothelial sprouting (red arrows), while SA-treated rings exhibit inhibition of outgrowth and vessel formation (n = 3 mice). Scale bars: 200 µm (top images) and 100 µm (bottom images). Data are presented as mean ± SEM. Statistical analysis: one-way ANOVA with Tukey's post hoc test. Significance: *$P < 0.05$. Exact $P$ values for the statistical comparisons are shown in Appendix Table S3. Source data are available online for this figure.

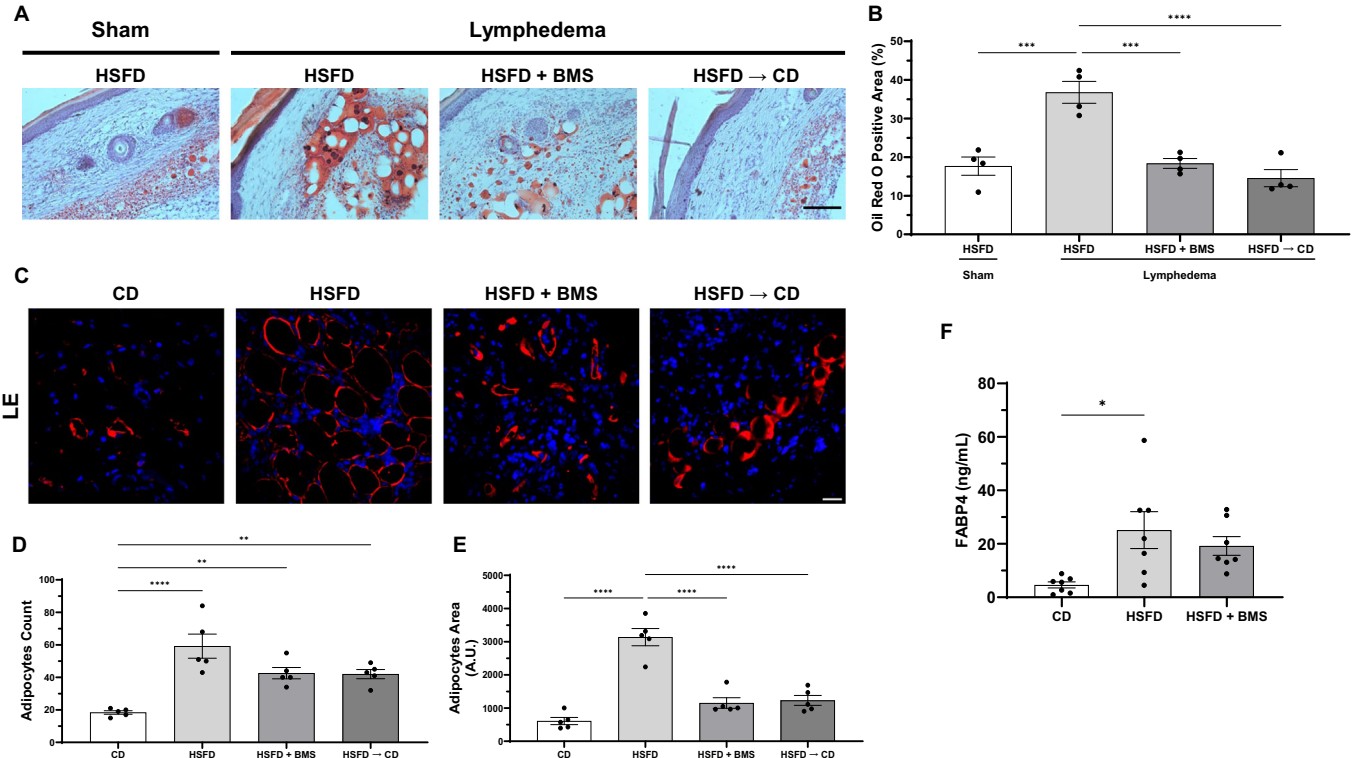

**Figure EV4. FABP4 inhibition or dietary transition reduces lipid accumulation and adipocyte expansion in lymphedematous tissue.**

(A, B) Oil Red O staining of tail tissue from sham and lymphedema (LE) mice maintained on a high saturated fat diet (HSFD) or treated with the FABP4 inhibitor BMS-309403 (HSFD + BMS) or switched to a control diet (HSFD → CD) after lymphatic injury. (A) Representative images showing lipid accumulation (red) in lymphedematous tails. Scale bars: 100 μm. (B) Quantification of Oil Red O positive area reveals increased lipid deposition in lymphedema tissue under HSFD, which is significantly reduced by BMS treatment or dietary transition ($n = 4$ mice). (C–E) Immunofluorescence staining for FABP4 (red) in tail tissue from CD, HSFD-, HSFD + BMS-, or HSFD → CD-fed lymphedema mice. DAPI (blue) marks nuclei. (C) Representative images showing increased FABP4 positive adipocytes in HSFD-fed mice. Scale bars: 50 μm. (D) Quantification of FABP4 positive adipocyte number. (E) Quantification of FABP4 positive adipocyte area ($n = 4$ mice). (F) Circulating FABP4 levels measured in mice with lymphedema, showing increased plasma FABP4 in HSFD-fed mice and partial reduction with BMS treatment ($n = 7$ mice). Data are presented as mean ± SEM. Statistical analysis: one-way ANOVA with Tukey's post hoc test. Significance: $*P < 0.05$, $**P < 0.01$, $***P < 0.001$, $****P < 0.0001$. Exact $P$ values for the statistical comparisons are shown in Appendix Table S3. HSFD high saturated fat diet, CD control diet, FABP4 fatty acid-binding protein 4, BMS BMS-309403, DAPI 4′,6-diamidino-2-phenylindole. Source data are available online for this figure.

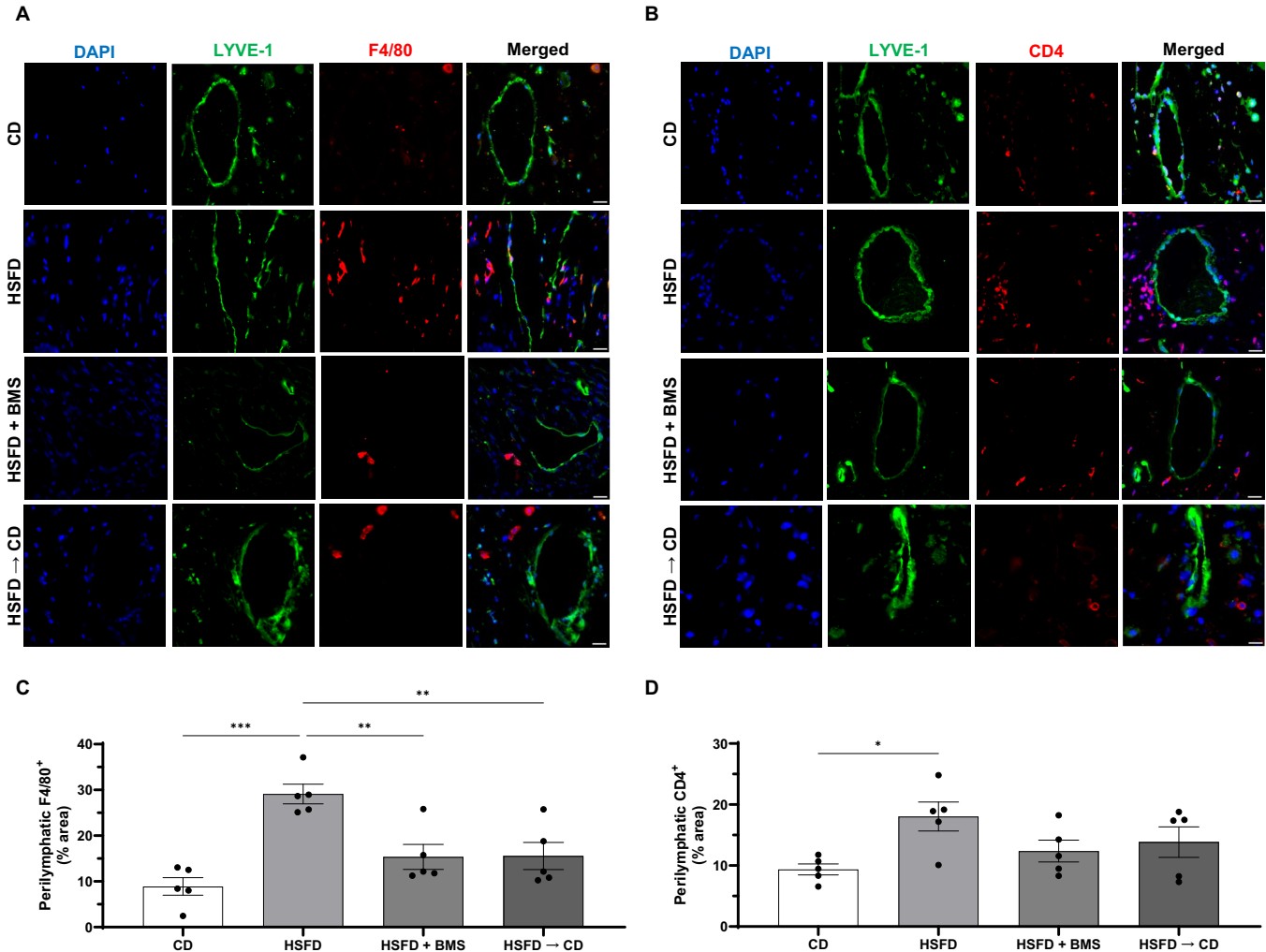

**Figure EV5. FABP4 inhibition and diet transition reduce immune cell accumulation in lymphedematous tissue.**

Representative immunohistochemistry images showing (**A**) F4/80 positive macrophages and (**B**) CD4 positive T cells in the perilymphatic region of tail tissue from mice with lymphedema fed a high saturated fat diet (HSFD), treated with the FABP4 inhibitor BMS-309403 (HSFD + BMS), or switched to a control diet after lymphatic injury (HSFD → CD). Scale bars: 50 μm. Quantification of immune cell infiltration for (**C**) F4/80 positive macrophages and (**D**) CD4 positive T cells (*n* = 5 mice). HSFD-fed mice exhibited significantly higher perilymphatic immune cell infiltration, which was attenuated by FABP4 inhibition or dietary transition. Data are presented as mean ± SEM. Statistical analysis: one-way ANOVA with Tukey's post hoc test. Significance: *$P < 0.05$, **$P < 0.01$, ***$P < 0.001$. Exact *P* values for the statistical comparisons are shown in Appendix Table S3. HSFD high saturated fat diet, CD control diet, FABP4 fatty acid-binding protein 4. Source data are available online for this figure.

