## [Peer Review File · EMBO Molecular Medicine]

Saturated fatty acids induce lipotoxicity in lymphatic endothelial cells contributing to lymphedema

Karina Gomes, Jacob Korodimas, Emily Liu, Nirav Patel, Xiaoyan Yang, Susan Goruk, Jaqueline Munhoz, Catherine Field, and Spencer Gibson

Corresponding author: Spencer Gibson (sgibson2@ualberta.ca)

Review Timeline:

Submission Date:	29th Jan 25
Editorial Decision:	27th Feb 25
Revision Received:	23rd Jun 25
Editorial Decision:	10th Jul 25
Revision Received:	15th Jul 25
Accepted:	21st Jul 25

Editor: Zeljko Durdevic

Transaction Report:

27th Feb 2025

Dear Dr. Gibson,

Thank you for the submission of your manuscript to EMBO Molecular Medicine. We have now received feedback from the three reviewers who agreed to evaluate your manuscript. All three referees recognize interest of the study but also raise important and partially overlapping concerns that should be addressed in a major revision. If you would like to discuss further the points raised by the referees, I am available to do so via email or video. Let me know if you are interested in this option.

We would welcome the submission of a revised version within three months for further consideration. Please let us know if you require longer to complete the revision.

I look forward to receiving your revised manuscript.

Yours sincerely,

Zeljko Durdevic

We require:

- 1) A .docx formatted version of the manuscript text (including legends for main figures, EV figures and tables). Please make sure that the changes are highlighted to be clearly visible.
- 2) Individual production quality figure files as .eps, .tif, .jpg (one file per figure). For guidance, download the 'Figure Guide PDF': (<https://www.embopress.org/page/journal/17574684/authorguide#figureformat>).
- 3) A .docx formatted letter INCLUDING the reviewers' reports and your detailed point-by-point responses to their comments. As part of the EMBO Press transparent editorial process, the point-by-point response is part of the Review Process File (RPF), which will be published alongside your paper.
- 4) A complete author checklist, which you can download from our author guidelines (<https://www.embopress.org/page/journal/17574684/authorguide#submissionofrevisions>). Please insert information in the checklist that is also reflected in the manuscript. The completed author checklist will also be part of the RPF.
- 5) Please note that all corresponding authors are required to supply an ORCID ID for their name upon submission of a revised manuscript.
- 6) It is mandatory to include a 'Data Availability' section after the Materials and Methods. Before submitting your revision, primary datasets produced in this study need to be deposited in an appropriate public database, and the accession numbers and

database listed under 'Data Availability'. Please remember to provide a reviewer password if the datasets are not yet public (see <https://www.embopress.org/page/journal/17574684/authorguide#dataavailability>).

12) Author contributions: You will be asked to provide CRediT (Contributor Role Taxonomy) terms in the submission system. These replace a narrative author contribution section in the manuscript.

13) A Conflict of Interest statement should be provided in the main text.

14) Every published paper now includes a 'Synopsis' to further enhance discoverability. Synopses are displayed on the journal webpage and are freely accessible to all readers. They include a short stand first (maximum of 300 characters, including space) as well as 2-5 one-sentences bullet points that summarizes the paper. Please write the bullet points to summarize the key NEW findings. They should be designed to be complementary to the abstract - i.e. not repeat the same text. We encourage inclusion of key acronyms and quantitative information (maximum of 30 words / bullet point). Please use the passive voice. Please attach

these in a separate file or send them by email, we will incorporate them accordingly.

15) Include a Reagents and Tools Table as part of the Methods section, which can be downloaded from our author guidelines (<https://www.embopress.org/page/journal/17574684/authorguide#structuredmethods>)

***** Reviewer's comments *****

Referee #1 (Comments on Novelty/Model System for Author):

Well established model of lymphedema in a mouse tail is used, and also human lymphatic endothelial cells in vitro (LECs). These are appropriate model systems for this work.

Referee #1 (Remarks for Author):

Gomes et al. examined the role of saturated fatty acids (SFAs) in secondary lymphedema. They show that SFAs, and specifically stearic acid, increases oxidative stress, ER stress and apoptosis of LECs. In a mouse model, diet high in saturated fat led to prolonged tissue swelling after surgical induction of lymphedema. Study identifies FABP4 as a potential therapeutic target in lymphedema and suggests that dietary interventions may help prevent or diminish lymphedema associated with surgery. This study presents novel and interesting findings. Weaknesses that need to be addressed are as follows:

1. Fig. 1: For more robust data, number of samples in control, healthy condition should be increased.
2. Fig. 2D, 4A, 5C, 7C: Images of immunofluorescent stains are too small and it is difficult to evaluate the results in the context of tissue. Larger areas should be shown with more lymphatics in view, as well as staining at higher magnification. LYVE-1 also stains macrophages what is clearly seen in many images (for example Fig. 2D, chow diet, lymphedema). Another marker should be used to exclude macrophages and make sure that LECs are being analyzed.
3. Fig. 2E, F, G: It is not clear what was measured. Please clarify what was quantified. Absolute numbers should be shown as well, and it needs to be described how many vessels were analyzed or total area analyzed per sample.
4. What is the evidence showing that LECs in the skin are exposed to fatty acids in vivo? This is an important point, since the study examines direct effects of SFAs on LECs. In vivo effects of SFAs on LECs may be indirect. What is the physiological concentration of SFAs in the tissues (skin)? Please address.
5. Fig. 3A, B: please show images of cells in the supplement. Were ECs sparse or confluent in the experiments shown in Fig. 3? It is important that all in vitro experiments be performed on confluent ECs, since ECs are not found as single cells in vivo. Cell contact or lack of it will greatly influence the results, since LECs will be more susceptible to various stressors in the absence of cell-cell contacts.
6. To compare effects on LECs vs. BECs, HDMECs should be used in all experiments instead or in addition to HUVEC.
7. Fig. 5A: Interesting data is shown that switch from HSFD to CD can prevent persistent edema i.e. help resolve lymphedema faster. However, authors indicate that the change in diet reverses lymphedema, but based on the experimental design where the diet had been changed before edema has fully developed, it is not reversed, it is prevented. It would be reversed if CD is started later, from day 20 on. How is the endpoint decided? It would be informative to see longer kinetics.
8. Fig. 6B: It is not clear what is shown here, since all three conditions show colocalization of calreticulin and FABP4. Again, in vitro studies should be done using confluent LECs, since LECs are not present in vivo as single cells.
9. Fig. EV7: It is unexpected that ECs formed colonies, since colony formation is a hallmark of transformed cells. Endothelial cell identity should be validated by staining with appropriate markers and bright field images of cultured ECs should be shown.

Referee #2 (Remarks for Author):

This is a highly interesting study that identifies SFAs as a potential driver of microenvironmental stress in lymphedema tissue, leading to ER stress, oxidative stress, and apoptosis. The study demonstrates that an HSFD diet prolongs lymphedema in a mouse model. Moreover, this phenotype was reversed by FABP4 inhibition, suggesting potential therapeutic strategies. Importantly, the PUFA/SFA ratio may serve as a potential biomarker for lymphedema risk and could guide dietary adaptations in patients.

Overall, this study is well-structured and holds crucial importance for patients with lymphedema. However, certain aspects of its presentation make it difficult to follow. Greater precision is needed to strengthen the message it conveys.

Figure 1 : The analysis of human data should be more accurate. The authors compared mildly overweight individuals (BMI 26.7) to obese lymphedema patients (BMI 33.6 > 30). It would be more informative to see analyses comparing individuals with the same BMI.

A separate graph and a clearer description should be provided for cancer-related lymphedema (likely lower limb LO) and non-cancer-related LO (chronic venous insufficiency in obese patients?).

Additionally, a table with PUFA, TG, and FFA dosages should be included.

Based on the presented data, the authors cannot conclusively state that "an imbalance in fatty acid composition may play a role in the pathophysiology of lymphedema, independent of obesity."

Figure 2 : It is surprising that there is no significant difference in mouse body weight between the CD and HFD groups (or are the statistics missing in EV3?).

The panels in Figure 2D are too small. The authors should provide larger panels, possibly with only merged images.

The quantifications do not match the provided images:

- No immunostaining of XBP1 (red) is shown in lymphatic vessels (green).
- No immunostaining of CHOP (red) is shown in lymphatic vessels (green), even in the HSFD group, where the green staining does not appear to represent lymphatics.
- TUNEL staining marks single cells surrounding lymphatics, which seem to resemble macrophages rather than lymphatic cells.

The authors should provide better staining (or alternative markers) and lower magnification.

What are the perilymphatic cells? Are they immune cells or adipocytes? This should be clarified.

Figure 3 : Mitox staining images should be included. A full-panel Western blot should be provided as supplementary data, along with the quantification of lipid droplets in EV9.

Figure 5 : The authors claim that dietary intervention significantly reduced tail swelling in the HSFD→CD group; however, no statistical analysis is provided to support this result.

The representative images are not convincing. For example, there is no visible 8-OHdG staining on lymphatics in the HSFD group.

Figure EV13 : It is difficult to believe that there is no expression of FABP4 in HUVECs, given that numerous studies report strong expression of this protein. While lower expression may be possible, a complete absence seems unlikely.

RT-qPCR data should be provided to reinforce this finding.

Additionally, an FABP4 knockdown experiment would confirm its role in lymphatic function.

Figure 7 : How did the authors verify FABP4 inhibition? Were circulating levels measured, as was done in patients?

Additional Discussion Points should be provided. While FABP4 inhibition appears beneficial for obese or diabetic patients, its impact on normal weight individuals remains unclear and could lead to metabolic and inflammatory imbalances. Further studies are needed to better understand these potential side effects. FABP4 plays a key role in fatty acid transport and metabolism in adipocytes and macrophages. Its inhibition could disrupt normal lipid regulation, leading to abnormal fatty acid accumulation, potentially affecting the liver and other metabolic tissues. Also, inhibiting FABP4 could increase circulating fatty acids, contributing to elevated triglycerides and atherogenic lipoproteins. The authors should discuss these aspects in greater detail. In addition, there are some non-answered questions: Why does HFD not increase lymphedema incidence, even though obesity is the most significant risk factor for lymphedema development? There is only a significant difference in linoleic acid levels—can this alone explain the observed phenotype? What is the linoleic acid level in the model of lymphedema?

Minor points:

The figures require editing. The font sizes in the graph legends vary too much, and some are too small. The image panels are too small, making the data difficult to follow. All in vitro survival assays are based on the Trypan Blue assay, yet no images are provided.

Referee #3 (Comments on Novelty/Model System for Author):

Indicated inadequate above as there is no cell-specificity demonstrated of any of the in vivo findings and a host of other systemic reasons for the outcomes observed.

Referee #3 (Remarks for Author):

Gomes and colleagues present a study in which stearic acid appears to exacerbate lymphedema via an ER stress mechanism in lymphatic endothelial cells. Lymphedema is underappreciated and the underlying etiology of the condition is still open for exploration. The work builds off of past studies from the lab and the field examining cell stress and the impact of diet versus obesity on lymphatic function and lymphedema. The overall conclusions that saturated fatty acids make things worse for lymphatic endothelial cells and lymphedema comes through clearly (and the title is thus supported), but the methodology and results fail to support the more specific mechanistic conclusions that the authors have made notably when translating from in vitro to in vivo.

Demonstrating fatty acid changes as ratios raises the immediate concern and question over which specific species and why is it necessary to report the data this way? Would 1E be correlated with any individual FA species? SFA alone? PUFA alone? What about differences in MUFA or UFA? This lead in data feels to be really pushing finding something to base the rest of the work on.

With FAs as a ratio, is it because the numerator is smaller or denominator bigger? The authors graph omega-6 PUFA linoleic acid and say that it is lower (see next concern), so the numerator is smaller... yet later when discussing EV11 it states that PUFA cannot rescue the SA driven effects. This makes it seem that the ratio, and effects, are driven by an increase in SFAs rather than a decrease in PUFA.

The statistical tests performed on the individual FA species are incorrect in EV2 and should be ANOVA to test if condition cause a change with a posthoc analysis. It does not look like 'lymphedema' would be a significant determinant of any changes, though it is possible that some are up and some are down. In the test saying "predominant" is incorrect, but rather "only LA was significantly lower" could get around this.

The in vivo FABP4 elevation could be from a variety of cell types, notably adipocytes and macrophages which are high expressers and clearly involved in the pathogenesis. While the patient data is supportive, it's not a great way to lead the figure. Better to show its involvement (and levels!) in mice, then say, 'and it's higher in patients!'

To the above point, the mouse studies with BMS is not lymphatic specific making it highly likely that inhibition in the other cell types is as likely a driver. Impact on adipocyte number, size, mac infiltration/phenotype?

The comparisons between patient groups in the Results are vague and not supported by any demonstrated statistical analysis between study groups in the table (needs stats).

That the HSFD is the only diet with a markedly different PUFA:SFA ratio, so this makes better sense to say (1) we employed this diet hoping to push the ratio to match humans and (2) that it worked. It's not study result. All of the HFD findings, since there are few, could be summed up here that, because we didn't push the ratio (or decrease PUFA or elevate SFA - however the above points are ironed out), that it was not longer employed. All the HFD data could then be removed.

The colocalization imaging experiments need to be explained in detail in the methods. The authors draw a lot of conclusions from these images and I simply do not see it. Why are lymphatics the only cells exhibiting these changes? Surely there is some xbp1s in other cells or TUNEL in other cells.

Similarly, conjugation of the FFAs for in vitro experiments is critical for batch to batch and species to species reproducibility. This is not described.

The authors report that their previous work identified OHdG as up in lymphedema, but here it is not raised in the other groups - only the LE+HSFD.

The diet transition time point is oddly chosen as there is no difference in volume yet at this time as per the earlier figure. It does suggest that reducing SFA intake can ameliorate, but this is an overall lower FA diet so not surprising that all adipose mass is reduced.

Resolution in the tail model dependent on wound healing and fluid transport/lymphangiogenesis across the wound so this should be assessed. Also the authors only examine lymphatics by immune in the tail region? Would other lymphatic beds also show this since the interventions are often systemic or are these changes in lymphedema on a high SFA diet or with inhibitors?

Minor: "...insufficient to induce obesity" - how is this defined

Minor: in the abstract and intro it should state the source tissue for where a lower PUFA:SFA was identified. Likewise FABP4 in patients.

Minor: saying that SFA causes these effects throughout is fine when specifically stating the outcome of the experiment, but

because PA does cause the effects as well, in likely addition to other saturated fats not tested, it should not be made as a generalization in introducing or discussing the work.

Author Responses to Reviewer Comments

Referee #1:

1. Fig. 1: For more robust data, number of samples in control, healthy condition should be increased.

Response: We thank the reviewer for the insightful comments and for emphasizing the importance of strengthening the control group. To increase the robustness of our control cohort, we added 15 breast cancer patients who were monitored for five years following cancer treatment and did not develop lymphedema during this follow-up period. This group provides a clinically relevant comparison, as most of the lymphedema samples in our study were from breast cancer-related lymphedema patients. With the inclusion of these breast cancer controls, we reassessed the PUFA/SFA ratio across lipid classes, and we found that differences between PUFA/SFA were still significant for triglycerides and cholesterol esters. The difference was no longer statistically significant for free fatty acids (the smallest fraction in plasma) due to breast cancer controls having a lower PUFA/SFA compared to healthy controls. In addition, the phospholipid PUFA/SFA was significantly lower in lymphedema when compared to breast cancer controls. These updated findings are now presented in Fig. 1, EV1, Appendix Table S1, and Appendix Figures S1 to S4.

2. Fig. 2D, 4A, 5C, 7C: Images of immunofluorescent stains are too small and it is difficult to evaluate the results in the context of tissue. Larger areas should be shown with more lymphatics in view, as well as staining at higher magnification. LYVE-1 also stains macrophages what is clearly seen in many images (for example Fig. 2D, chow diet, lymphedema). Another marker should be used to exclude macrophages and make sure that LECs are being analyzed.

Response: To improve clarity and provide better spatial context, we increased the size of immunofluorescent images (Fig. 2C, 4A, 5C, and 7C). New representative images are now presented to better show the lymphatic regions. We also moved Fig. 2C HFD images to EV2 to allow for more space for the enlarged images. In Fig. 4A, HFD images were moved to Appendix S12. In Fig. 5C, images of CD were moved to Appendix S15A. In Fig. 7C, CD + PBS and CD + BMS images were moved to Appendix S21. In addition, 8-OHdG staining in Fig. 7C was moved to Appendix S20.

Given that LYVE-1 can label subsets of macrophages, we also relied on morphological criteria such as vessel shape, lumen presence, and structural organization to distinguish true lymphatic vessels from individual LYVE-1⁺ macrophages. Additionally, in select experiments, we used F4/80 as a secondary macrophage marker to validate the identity of LYVE-1⁺ cells. We found

that macrophages are indeed increased in HSFD lymphedema tissue, but localized in different regions compared to the LYVE-1⁺ lymphatic structures (revised EV5). Furthermore, FABP4 inhibitor and dietary transition from HSFD to CD significantly reduced macrophages in the tail lymphedema tissue presented in EV5.

3. Fig. 2E, F, G: It is not clear what was measured. Please clarify what was quantified. Absolute numbers should be shown as well, and it needs to be described how many vessels were analyzed or total area analyzed per sample.

Response: We thank the reviewer for the opportunity to clarify our quantification methods in Figures 2E, F, and G. In these panels, we measured the colocalization of the selected markers with LYVE-1. All the colocalization analyses in this study were performed within the LYVE-1⁺ vessel ROIs. Given that LYVE-1 can also label subsets of macrophages, additional morphological features such as vessel shape and lumen presence were used to distinguish true lymphatic structures. Colocalization of each marker with LYVE-1 was assessed using the Pearson's correlation coefficient, which is now explicitly indicated on the Y-axis of the graphs. A minimum of five high-powered fields and five lymphatic vessels per animal were analyzed. In addition, we have revised the Materials and Methods section to provide greater clarity on how this analysis was done. These methodological details have been added to ensure reproducibility and transparency.

4. What is the evidence showing that LECs in the skin are exposed to fatty acids in vivo? This is an important point, since the study examines direct effects of SFAs on LECs. In vivo effects of SFAs on LECs may be indirect. What is the physiological concentration of SFAs in the tissues (skin)? Please address.

Response: We thank the reviewer for this important question regarding the in vivo exposure of LECs to fatty acids, which is central to the interpretation of our study. We used Oil Red O (ORO) staining to visualize neutral lipids, including triglycerides, cholesteryl esters, and lipoproteins in the tail tissue after the various diets and treatments. We found that HSFD showed higher levels of ORO staining compared to tail tissue with HFD and CD. This lipid accumulation indicates that the fatty acids are entering the tail tissue and being accumulated. In addition, ORO staining was reduced following FABP4 inhibitor treatment and dietary switch from HSFD to CD (revised EV 4). These findings support the conclusion that fatty acids and other lipids accumulate in the lymphedema tissue, making direct exposure of local cells (including LECs) to fatty acids plausible. While this evidence is consistent with in vivo lipid exposure, we acknowledge that definitive identification of fatty acid species and their cell-type-specific localization remains to be determined. To address this, we are planning future studies using lipidomic profiling by gas

chromatography–mass spectrometry to quantify and characterize fatty acids in lymphedema tissue and isolated cell populations.

5. Fig. 3A, B: Please show images of cells in the supplement. Were ECs sparse or confluent in the experiments shown in Fig. 3? It is important that all in vitro experiments be performed on confluent ECs, since ECs are not found as single cells in vivo. Cell to cell contact or lack of it will greatly influence the results, since LECs will be more susceptible to various stressors in the absence of cell-cell contacts.

Response: All drug treatments were performed consistently when cell cultures reached approximately 70–80% confluency. That corresponds to a subconfluent state that allows for initial cell–cell contact while preventing overgrowth in untreated control plates during the treatment period, which could otherwise lead to nutrient depletion and loss of viability. We have included representative bright field phase-contrast images of cell morphology and confluency at the time of treatment (Appendix S23). We have also clarified this experimental detail in the revised Materials and Methods section to enhance transparency and reproducibility.

6. To compare effects on LECs vs. BECs, HDMECs should be used in all experiments instead or in addition to HUVEC.

Response: We have now included HDMECs in the analysis. We found that these cells were resistant to stearic acid-induced cell death, and stearic acid did not significantly increase ROS production in these cells, in contrast to the pronounced effects observed in HDLECs (Fig. 3E and Appendix S9). In addition, analysis of FABP4 protein expression showed that HDMECs express FABP4 at levels comparable to HUVECs but lower than HDLECs (Appendix S10), which may partly explain the differential sensitivity to stearic acid–induced stress. These additional data support the concept that LECs are more susceptible to saturated fatty acid–induced cellular stress compared to blood endothelial cells.

7. Fig. 5A: Interesting data is shown that switch from HSFD to CD can prevent persistent edema i.e. help resolve lymphedema faster. However, authors indicate that the change in diet reverses lymphedema, but based on the experimental design where the diet had been changed before edema has fully developed, it is not reversed, it is prevented. It would be reversed if CD is started later, from day 20 on. How is the endpoint decided? It would be informative to see longer kinetics.

Response: We agree that the distinction between prevention and reversal of lymphedema is important for accurate interpretation of our findings. We have revised the text to clarify that. In this model, the dietary switch from HSFD to CD at the time of surgery prevented the delayed

recovery in tail swelling typically observed under continuous HSFD feeding. Our experimental design aimed to test whether dietary intervention at the time of lymphatic injury could alter the course of disease progression. The choice of 28 days post-surgery as the experimental endpoint was based on preliminary studies using CD-fed mice, which showed a consistent ~30% reduction in tail swelling by this time point, marking the onset of the recovery phase. We agree that evaluating longer time points and initiating dietary changes at later stages would provide valuable insight into the potential for true reversal of established lymphedema. These kinetic studies will be the focus of future investigations.

8. Fig. 6B: It is not clear what is shown here, since all three conditions show colocalization of calreticulin and FABP4. Again, in vitro studies should be done using confluent LECs, since LECs are not present in vivo as single cells.

Response: This panel shows that FABP4 colocalizes with calreticulin, an ER marker, under all treatment conditions, indicating that a substantial fraction of FABP4 localizes to the ER. As with our other in vitro experiments, these evaluations were conducted at a sub-confluent state (70-80%), which allows for cell-cell contact while avoiding overgrowth. We have now included new representative fluorescence images at lower magnification, showing the confluency state of the cells at the time of the staining. This is presented in Fig. 6D. The degree of colocalization between FABP4 and calreticulin was quantified using Pearson's correlation coefficient, and these results are now presented in Fig. 6E. We have also clarified these experimental details in the revised Materials and Methods section.

9. Fig. EV7: It is unexpected that ECs formed colonies, since colony formation is a hallmark of transformed cells. Endothelial cell identity should be validated by staining with appropriate markers and bright field images of cultured ECs should be shown.

Response: We appreciate the reviewer's comment and understand the concern regarding colony formation, which is more commonly associated with transformed cells. However, our approach was based on a previously published protocol demonstrating colony-forming ability of both blood and lymphatic endothelial cells under specific culture conditions (Angiogenesis 2014 Apr;17(2):419-27). This assay has been used to assess the clonogenic potential and proliferative capacity of primary endothelial cells without implying transformation. To preserve endothelial identity and avoid phenotypic drift, we used primary LECs at low passage numbers (up to P8) for all experiments.

Referee #2:

Figure 1: The analysis of human data should be more accurate. The authors compared mildly overweight individuals (BMI 26.7) to obese lymphedema patients (BMI 33.6 > 30). It would be more informative to see analyses comparing individuals with the same BMI.

Response: We thank the reviewer for this important comment regarding BMI in our human cohort analysis. We have now expanded our control group by including 15 breast cancer control patients who were monitored for five years following cancer treatment and did not develop lymphedema during this follow-up period. This expanded cohort provided a broader BMI distribution, enabling comparisons between individuals with similar BMI ranges in both the lymphedema and control groups (Fig. 1, EV1, Appendix Table S1, Appendix Figure S1). Upon reanalysis, we observed that the low PUFA/SFA ratios in lymphedema patients were not correlated with BMI, suggesting that the altered lipid composition is not solely attributable to obesity. However, in the control group, triglyceride PUFA/SFA ratios showed a negative correlation with BMI, consistent with known metabolic effects of increased adiposity (Appendix Figure S2). These updates strengthen our interpretation that lymphatic dysfunction, rather than BMI alone, may contribute to altered lipid profiles in lymphedema.

A separate graph and a clearer description should be provided for cancer-related lymphedema (likely lower limb LO) and non-cancer-related LO (chronic venous insufficiency in obese patients?).

Response: As detailed in Appendix Table S1, the majority of lymphedema samples were cancer-related, including both upper limb (e.g., breast cancer) and lower limb (e.g., gynecological cancers, prostate cancer). In contrast, only five patients presented with non-cancer-related, obesity-associated lower limb lymphedema. Due to the limited number of non-cancer cases, separate statistical analysis was not feasible. Nonetheless, we have now clarified this patient stratification in Appendix Table S1 to improve transparency.

Additionally, a table with PUFA, TG, and FFA dosages should be included.

Response: Due to limitations in our GC-MS setup, we were unable to perform absolute quantification of fatty acids in the human plasma samples. Consequently, our analysis is based on the relative proportions of fatty acid species within each lipid class. This is described in the Materials and Methods section.

Based on the presented data, the authors cannot conclusively state that "an imbalance in fatty acid composition may play a role in the pathophysiology of lymphedema, independent of obesity."

Response: With the addition of new breast cancer control patients and stratification of PUFA/SFA ratios across BMI ranges, we now observe that lymphedema patients with lower BMI still exhibit significantly reduced PUFA/SFA ratios in both phospholipids and triglycerides fractions compared to controls (Appendix Figure S2). These findings strengthen our original interpretation that altered fatty acid composition, rather than BMI alone, may contribute to lymphatic dysfunction in lymphedema. However, we fully acknowledge that obesity remains an important contributing factor that can exacerbate lymphedema severity through metabolic, inflammatory, and mechanical pathways. We have revised the manuscript to reflect a more balanced interpretation, emphasizing that both lipid-altered proportions and obesity likely play interacting roles in lymphedema progression.

Figure 2: It is surprising that there is no significant difference in mouse body weight between the CD and HFD groups (or are the statistics missing in EV3?).

Response: We thank the reviewer for this important observation and apologize for the oversight. We have now included statistical analyses, which show that a significant difference in body weight between CD and HFD/HSFD groups is present only at the final week of the dietary intervention. The 8-week duration was intentionally chosen to avoid the onset of pronounced obesity-related metabolic changes, enabling us to isolate the impact of dietary fat composition on lymphatic outcomes. As expected, HFD- and HSFD-fed mice showed a modest (~16%) but statistically significant increase in body weight compared to CD-fed controls at the end of the dietary intervention. This level of weight gain is not considered obesity, particularly in female C57BL/6 mice, which are known to be resistant to diet-induced obesity over moderate time frames (J Diabetes Complications. 2021 Feb;35(2):107795). These updates are now reflected in the revised figure legend.

The panels in Figure 2D are too small. The authors should provide larger panels, possibly with only merged images.

Response: To address the reviewer's concern about data presentation, we have now increased the size of immunofluorescent images (Fig. 2C, 4A, 5C, and 7C). We also moved Fig. 2C HFD images to EV2 to allow for more space for the enlarged images. In Fig. 4A, HFD images were moved to Appendix S12. In Fig. 5C, images of CD were moved to Appendix S15A. In Fig. 7C, CD + PBS and CD + BMS images were moved to Appendix S21. In addition, 8-OHdG staining in Fig. 7C was moved to Appendix S20.

The quantifications do not match the provided images:

- **No immunostaining of XBP1 (red) is shown in lymphatic vessels (green).**
- **No immunostaining of CHOP (red) is shown in lymphatic vessels (green), even in the HSFD group, where the green staining does not appear to represent lymphatics.**
- **TUNEL staining marks single cells surrounding lymphatics, which seem to resemble macrophages rather than lymphatic cells.**

Response: We thank the reviewer for the opportunity to clarify and improve the presentation. In response, we have updated the relevant figures to enhance the quality of the representative images. For XBP1 and CHOP immunostaining, we revised the image panels to better highlight their localization within LYVE-1⁺ lymphatic vessels. In some HSFD samples, LYVE-1 morphology appears altered due to tissue remodeling or lymphatic dilation, but vessel identity was consistently confirmed using established criteria, including vessel shape, lumen presence, and continuity of LYVE-1 signal. For TUNEL staining, we quantified TUNEL⁺ nuclei within or immediately adjacent to LYVE-1⁺ vessels. While some perilymphatic TUNEL⁺ cells may represent macrophages, our prior work (Int J Mol Sci. 2024; 25:7828) and current staining patterns suggest the majority are lymphatic endothelial cells. To support this, we performed F4/80 immunostaining, which showed a distinct distribution of macrophages that did not overlap with LYVE-1⁺ structures (Figure EV5), reinforcing the specificity of our analysis. These updates are reflected in the revised figures and clarified in the Materials and Methods section for transparency and reproducibility.

The authors should provide better staining (or alternative markers) and lower magnification.

What are the perilymphatic cells?

Response: In our study, perilymphatic cells refer to cells located within a 100 μm radius surrounding LYVE-1⁺ lymphatic vessels. This region was selected to capture cells directly adjacent to the lymphatics, where apoptotic and inflammatory responses are most likely to occur. The perilymphatic space was defined and applied consistently across all samples for quantification. This approach is supported by prior studies (Nat Commun. 2018 May 17;9(1):1970; Front Aging. 2022 Apr 4;3:864860; Int J Obes. 2016 Jun 21;40(10):1582–1590).

While we acknowledge that LYVE-1 is not exclusive to lymphatic endothelial cells and may occasionally label subsets of macrophages, vessel identity was confirmed based on morphological features, including vessel shape, lumen presence, and continuous LYVE-1 staining. In addition, to distinguish lymphatics from macrophages, we performed F4/80

immunostaining, which revealed a non-overlapping distribution of macrophages, supporting the specificity of our perilymphatic analysis (see revised Appendix Figure EV5). We have clarified our perilymphatic analysis in the Materials and Methods section.

Are they immune cells or adipocytes? This should be clarified.

Response: We thank the reviewer for this important question. We performed immunostaining for macrophages (F4/80) and T cells (CD4) in lymphedema tail tissue and observed increased infiltration of both cell types in mice fed a HSFD (Appendix Figure EV5). In addition, we used Oil Red O staining as a surrogate marker and observed increased lipid accumulation consistent with adipocyte expansion in the lymphedematous tails of HSFD-fed mice (Figure EV4A). We also conducted FABP4 immunostaining, which supports an increase in mature adipocytes within the lymphedematous tissue under HSFD conditions (Figure EV4B). Although FABP4 is also expressed by other cells, it remains a widely used marker for adipocyte-rich regions, especially in inflamed or metabolically altered environments. These findings indicate that both immune cells and adipocytes accumulate in lymphedema tissue in response to HSFD.

Figure 3: Mitosox staining images should be included. A full-panel Western blot should be provided as supplementary data, along with the quantification of lipid droplets in EV9.

Response: The MitoSOX experiments were performed using flow cytometry, not immunostaining. Representative flow cytometry plots are included in Appendix Figures S9 and S10. In addition, full-panel Western blots have been provided as Source Data files. The quantification of lipid droplet accumulation in vitro has also been added and is presented in Appendix Figure S8.

Figure 5: The authors claim that dietary intervention significantly reduced tail swelling in the HSFD→CD group; however, no statistical analysis is provided to support this result.

Response: We thank the reviewer for pointing this out and apologize for the oversight. We have now included the appropriate statistical analyses in Figures 5A and 5B, which confirm that the reduction in tail swelling in the HSFD→CD group is statistically significant. These updates are now reflected in the revised figure and legend.

The representative images are not convincing. For example, there is no visible 8-OHDG staining on lymphatics in the HSFD group.

Response: We have revised the representative image in Figure 4A to more clearly show co-staining of 8-OHdG and LYVE-1 in the HSFd group. The updated figure better highlights oxidative DNA damage within lymphatic vessels.

Figure EV13: It is difficult to believe that there is no expression of FABP4 in HUVECs, given that numerous studies report strong expression of this protein. While lower expression may be possible, a complete absence seems unlikely.

Response: While FABP4 is expressed in HUVECs, its levels are markedly lower compared to HDLECs. To clarify this point, we re-exposed the Western blots, including overexposed membranes to visualize faint bands. The updated images now clearly show detectable FABP4 expression in both HUVECs and HDMECs, though expression remains significantly lower than in HDLECs (Appendix Figure S17).

We have included HDMECs (human dermal microvascular endothelial cells) in our analysis to provide a more physiologically relevant comparison to HDLECs, as both cell types are derived from microvascular beds of the skin. This allows for a more appropriate assessment of lineage- and tissue-specific differences in FABP4 expression and cellular responses to saturated fatty acids.

Additionally, an FABP4 knockdown experiment would confirm its role in lymphatic function.

Response: We thank the reviewer for this valuable experimental suggestion. In response, we performed FABP4 knockdown using siRNA in HDLECs and observed that stearic acid-induced cell death was significantly reduced in FABP4-deficient cells, supporting its role in mediating lipotoxicity in lymphatic endothelial cells (Figure 6F and G).

Figure 7: How did the authors verify FABP4 inhibition? Were circulating levels measured, as was done in patients?

Response: To verify FABP4 inhibition in vivo, we measured circulating FABP4 levels in mouse plasma and found that treatment with the FABP4 inhibitor attenuated the systemic FABP4 levels in HSFd-fed mice with lymphedema (Appendix Figure EV4F). Additionally, Oil Red O and FABP4 staining showed reduced lipid accumulation and adipocyte area in lymphedema tissue (Appendix EV4A–D), suggesting that FABP4 inhibition could limit lipid uptake and accumulation in adipose tissue within the lymphedema environment. Together, these results indicate that BMS inhibits FABP4 expression both systemically and locally, and suggest a role for FABP4 in lipid-driven tissue remodeling.

Additional Discussion Points should be provided. While FABP4 inhibition appears beneficial for obese or diabetic patients, its impact on normal weight individuals remains unclear and could lead to metabolic and inflammatory imbalances. Further studies are needed to better understand these potential side effects. FABP4 plays a key role in fatty acid transport and metabolism in adipocytes and macrophages. Its inhibition could disrupt normal lipid regulation, leading to abnormal fatty acid accumulation, potentially affecting the liver and other metabolic tissues. Also, inhibiting FABP4 could increase circulating fatty acids, contributing to elevated triglycerides and atherogenic lipoproteins. The authors should discuss these aspects in greater detail.

Response: We have expanded the Discussion to address potential metabolic risks of FABP4 inhibition, particularly in normal-weight individuals. While FABP4 inhibition may benefit obesity-associated lymphedema, it could also disrupt lipid metabolism and increase circulating fatty acids. These concerns are now discussed, emphasizing the need for further studies to evaluate long-term safety.

In addition, there are some non-answered questions: Why does HFD not increase lymphedema incidence, even though obesity is the most significant risk factor for lymphedema development?

Response: While obesity is a known risk factor for lymphedema, in our study, HFD alone did not impair recovery. However, we did observe higher initial swelling in HFD-fed mice compared to CD controls, suggesting a potential acute effect on lymphatic function. In contrast, HSFD-fed mice exhibited both greater swelling and delayed recovery, indicating that saturated fat content plays a more direct role in impairing lymphatic repair. This distinction is now discussed in the revised manuscript to highlight the need for further investigation into the independent effects of diet composition versus obesity on lymphedema pathogenesis.

There is only a significant difference in linoleic acid levels-can this alone explain the observed phenotype? What is the linoleic acid level in the model of lymphedema?

Response: With the inclusion of additional controls, we observed broader lipid alterations beyond linoleic acid reduction, including increased saturated fatty acids (SFAs) and decreased polyunsaturated fatty acids (PUFAs) across several lipid classes (phospholipids, triglycerides, and cholesterol esters), with linoleic acid being the primary affected PUFA (Figure EV1). Similar changes were seen in the mouse model, where HSFD-fed mice showed reduced PUFA/SFA ratios driven by increased stearic acid and decreased linoleic acid (Figure EV2B). These findings

suggest significant shifts in the proportions of PUFAs and SFAs in the plasma of both lymphedema patients and HSFD-fed mice with tail lymphedema.

Minor points:

The figures require editing. The font sizes in the graph legends vary too much, and some are too small. The image panels are too small, making the data difficult to follow. All in vitro survival assays are based on the Trypan Blue assay, yet no images are provided.

Response: We thank the reviewer for the suggestion. Font sizes and panel dimensions have been standardized for clarity, and representative Trypan Blue assay images are now included in Appendix Figure S23B.

Referee #3:

Demonstrating fatty acid changes as ratios raises the immediate concern and question over which specific species and why is it necessary to report the data this way? Would 1E be correlated with any individual FA species? SFA alone? PUFA alone? What about differences in MUFA or UFA? This lead in data feels to be really pushing finding something to base the rest of the work on.

Response: We thank the reviewer for this insightful comment. The PUFA/SFA ratio is a widely accepted metric for assessing lipid balance and its relevance to metabolic and inflammatory disorders (e.g., *Int J Mol Sci.* 2024;25(17):9288; *Am J Clin Nutr.* 2005;82(6):1178–84). In response to the reviewer's concern, we now provide the proportions of the major individual fatty acids across all major lipid classes in the revised Figure EV1 and Appendix Figure S4.

These updated data include an expanded control cohort of breast cancer patients who did not develop lymphedema within five years of treatment (Appendix Table S1). This allowed a more balanced comparison and revealed that the altered PUFA/SFA ratio is largely driven by reduced linoleic acid and increased SFAs. We also examined MUFA and total UFA levels, which showed no consistent differences aside from a modest shift in the free fatty acid fraction. These results are now reflected in the revised Results section and support the PUFA/SFA ratio as a meaningful indicator of fatty acid alterations and differences in composition in both patients and our mouse model.

With FAs as a ratio, is it because the numerator is smaller or denominator bigger? The authors graph omega-6 PUFA linoleic acid and say that it is lower (see next concern), so the numerator is smaller... yet later when discussing EV11 it states that PUFA cannot rescue the SA driven effects. This makes it seem that the ratio, and effects, are driven by an increase in SFAs rather than a decrease in PUFA.

Response: The reduced PUFA/SFA ratio observed in plasma reflects both a decrease in PUFA (primary linoleic acid) and an increase in SFA (primary palmitic acid), with the relative contribution varying by lipid fraction. The ratio captures this alteration in fatty acid composition rather than a unidirectional shift.

We acknowledge that plasma profiling and in vitro experiments assess different contexts: circulating lipids versus direct cellular effects. While linoleic acid levels were reduced in patient plasma, we initially did not observe a protective effect in vitro, likely due to technical limitations. We have since optimized our handling protocol by using nitrogen-flushed solutions to minimize fatty acid oxidation. This concern is especially pertinent to PUFAs, whose multiple double bonds make them prone to peroxidation in the presence of oxygen, thereby diminishing their stability and effective concentration in culture media. Under these improved conditions, we found that linoleic acid did mitigate stearic acid-induced cell death in HDLECs (Appendix

Figure S14), supporting its protective role. These methodological refinements and findings are now reflected in the revised Methods, Results, and Discussion sections.

It is important to note that while PUFA handling requires special precautions to prevent oxidative degradation, SFAs such as stearic acid and palmitic acid are structurally saturated and chemically stable under standard cell culture conditions. Their lack of double bonds renders them highly resistant to spontaneous oxidation, so nitrogen flushing is not typically necessary for their use in vitro. Therefore, the effects observed with SFA treatment are unlikely to be influenced by oxidative loss or degradation, in contrast to PUFAs, whose biological activity can be significantly underestimated if not properly stabilized (J Nutr. 2012 Mar;142(3):610S-613S; AAPS PharmSciTech. 2022 May 20;23(5):151).

The statistical tests performed on the individual FA species are incorrect in EV2 and should be ANOVA to test if condition cause a change with a posthoc analysis. It does not look like 'lymphedema' would be a significant determinant of any changes, though it is possible that some are up and some are down. In the test saying "predominant" is incorrect, but rather "only LA was significantly lower" could get around this.

Response: We thank the reviewer for this important observation. We have re-analyzed the fatty acid data in (please see the new Figure EV1) using one-way ANOVA with post hoc testing to better evaluate group differences. With the addition of new control samples, we now observe that linoleic acid is significantly reduced, and palmitic acid is significantly increased in the phospholipid fraction of plasma from lymphedema patients. Alterations in the phospholipid fraction are particularly meaningful because phospholipids are essential components of cellular membranes and reflect long-term or chronic adaptations in lipid metabolism. Unlike free fatty acids, which fluctuate with dietary intake or acute metabolic shifts, the fatty acid composition of phospholipids changes more slowly and provides a stable indication of sustained lipid remodeling. In addition, we found a significant increase in total SFAs within the triglyceride fraction. Although TGs primarily serve as an energy reservoir, elevated SFA content in this fraction may indicate altered lipid storage and mobilization, contributing to systemic metabolic stress. These updated analyses and their implications are now reflected in Figure EV1 and Appendix Figure S4 and the revised Results and Discussion sections.

The in vivo FABP4 elevation could be from a variety of cell types, notably adipocytes and macrophages which are high expressers and clearly involved in the pathogenesis. While the patient data is supportive, it's not a great way to lead the figure. Better to show its involvement (and levels!) in mice, then say, 'and it's higher in patients!'

Response: We appreciate the reviewer's thoughtful suggestion. We agree that multiple cell types—particularly adipocytes and macrophages—are well-known sources of circulating FABP4,

and we now acknowledge this in the revised text. Interestingly, recent evidence suggests that endothelial cells represent a major contributor to circulating FABP4 levels in mice (JCI Insight. 2023 Jul 24;8(14):e164642).

While we understand the rationale for reordering the data to begin with the mouse model, we have chosen to retain the current structure to preserve the logical progression of our findings, which were initially based on human patient observations and subsequently validated through mechanistic studies in mouse models and cell culture. We have now included plasma FABP4 measurements in HSF-fed lymphedema mice (Figure EV4F), which show elevated levels consistent with the patient data.

To the above point, the mouse studies with BMS is not lymphatic specific making it highly likely that inhibition in the other cell types is as likely a driver. Impact on adipocyte number, size, mac infiltration/phenotype?

Response: We agree with the reviewer that BMS may inhibit FABP4 in multiple cell types beyond lymphatic endothelial cells, including adipocytes and macrophages. To better understand these contributions, we conducted additional experiments, now included in Figures EV4 and EV5. Specifically, we found that FABP4 inhibition reduced adipocyte area and lipid accumulation in lymphedematous tails (Oil Red O staining and FABP4 immunostaining, EV4 A–D), and also decreased macrophage and T cell infiltration (F4/80 and CD4 staining, EV5 A–B). These results suggest that FABP4 inhibition affects both adipose expansion and immune cell recruitment, which likely contribute to the observed improvements in tissue remodeling.

While these findings support a broader impact of FABP4 inhibition in the lymphedema model, they also highlight the complexity of its role across different cell types. We have updated the Discussion section to acknowledge that future studies using lymphatic endothelial cell-specific FABP4 knockout models will be essential to delineate cell-intrinsic effects from systemic contributions.

The comparisons between patient groups in the Results are vague and not supported by any demonstrated statistical analysis between study groups in the table (needs stats).

Response: We have now added a control group of breast cancer patients without lymphedema and performed appropriate statistical comparisons between all study groups. These 15 breast cancer patients were monitored for five years following cancer treatment and did not develop lymphedema during this follow-up period. This group provides a clinically relevant comparison, as most of the lymphedema samples in our study were from breast cancer-related lymphedema patients. With the inclusion of these breast cancer controls, we reassessed the PUFA/SFA ratio across lipid classes. This analysis confirms that the PUFA/SFA ratio is significantly reduced in

most lipid fractions in lymphedema patients (revised Fig. 1, EV1, Appendix Table S1, and Appendix Figures S1 to S4).

That the HSF is the only diet with a markedly different PUFA:SFA ratio, so this makes better sense to say (1) we employed this diet hoping to push the ratio to match humans and (2) that it worked. It's not study result. All of the HFD findings, since there are few, could be summed up here that, because we didn't push the ratio (or decrease PUFA or elevate SFA - however the above points are ironed out), that it was not longer employed. All the HFD data could then be removed.

Response: We thank the reviewer for the helpful suggestion. We have streamlined the presentation by moving the HFD data to Figure EV2 and Appendix Figures S12 and S19.

The colocalization imaging experiments need to be explained in detail in the methods. The authors draw a lot of conclusions from these images and I simply do not see it. Why are lymphatics the only cells exhibiting these changes? Surely there is some xbp1s in other cells or TUNEL in other cells.

Response: We have now clarified in the Materials and Methods that colocalization analysis was performed using Pearson's correlation coefficient to quantify signal overlap between LYVE-1 and markers such as sXBP1 and TUNEL. To evaluate whether these markers were exclusive to lymphatic endothelial cells, we also conducted a perilymphatic analysis (e.g., Figure 2H–J). In this approach, we quantified marker-positive cells within a defined 100 μm radius surrounding LYVE-1⁺ lymphatic vessels to capture adjacent stromal and immune cells. This analysis revealed that ER stress and apoptosis markers were not limited to lymphatic endothelial cells but were also present in neighboring cell populations. These methodological details and findings are now more clearly described in the revised Methods and Results sections.

Similarly, conjugation of the FFAs for in vitro experiments is critical for batch to batch and species to species reproducibility. This is not described.

Response: We thank the reviewer for pointing this out. We have now revised the Materials and Methods section to include a detailed description of our fatty acid conjugation protocol. Saturated fatty acids (stearic acid and palmitic acid) were conjugated to fatty acid-free BSA following established protocols to ensure solubility and bioavailability. Polyunsaturated linoleic acid was handled under nitrogen to prevent oxidation and similarly conjugated to BSA. These standardized procedures were used across all experiments to ensure batch-to-batch and species-to-species reproducibility. Additionally, we used sample stocks to further minimize variability.

The authors report that their previous work identified OHdG as up in lymphedema, but here it is not raised in the other groups - only the LE+HSFD.

Response: We thank the reviewer for the opportunity to clarify this point. Across all IHC experiments in this study, the figures are intended to illustrate relative differences in marker expression between experimental groups, rather than indicate complete absence of signal in any condition. Fluorescence image acquisition was performed using fixed settings across all samples: exposure time and laser power were calibrated based on the group with the highest expected fluorescence, and these parameters were applied uniformly to ensure consistency and eliminate post-acquisition bias. As a result, representative images from groups with lower marker expression may appear dimmer, accurately reflecting biologically relevant differences.

Specifically regarding 8-OHdG staining, our previous work demonstrated that oxidative stress markers increase during the early phase of lymphedema progression (12 days post-injury) but decline as swelling resolves in CD-fed mice (Int J Mol Sci. 2024;25:7828). In contrast, our current data show sustained elevation of oxidative stress at a later stage (28 days post-injury) in HSFD-fed mice, indicating a prolonged oxidative and inflammatory response likely driven by dietary lipid alterations.

The diet transition time point is oddly chosen as there is no difference in volume yet at this time as per the earlier figure. It does suggest that reducing SFA intake can ameliorate, but this is an overall lower FA diet so not surprising that all adipose mass is reduced.

Response: The diet switch was timed shortly after the surgery to test whether early intervention could prevent or reduce swelling. While no volume difference was present at the transition point, HSFD→CD mice showed reduced swelling later, suggesting a preventive effect. As noted in the revised Discussion, future studies will assess the impact of dietary changes introduced after swelling onset to better define therapeutic timing.

Resolution in the tail model dependent on wound healing and fluid transport/lymphangiogenesis across the wound so this should be assessed. Also the authors only examine lymphatics by immune in the tail region? Would other lymphatic beds also show this, since the interventions are often systemic or are these changes in lymphedema on a high SFA diet or with inhibitors?

Response: We thank the reviewer for raising this important point. While we were unable to directly assess lymphangiogenesis in vivo in the tail model, we addressed this limitation by performing complementary in vitro and ex vivo assays that evaluate key cellular processes involved in lymphatic repair.

To examine lymphatic endothelial cell migration, we conducted a scratch wound assay using HDLECs and HDMECs. We have included HDMECs (human dermal microvascular endothelial cells) in our analysis to provide a more physiologically relevant comparison to HDLECs, as both cell types are derived from microvascular beds of the skin. Under control conditions, HDLECs significantly closed the wound area within 32 hours; however, this migratory response was markedly impaired by treatment with stearic acid. In contrast, HDMECs migration was less affected, and wounds closed regardless of stearic acid exposure (Figures EV3A–D). To further assess lymphatic vessel formation, we performed an ex vivo lymphatic ring assay using mouse thoracic duct segments embedded in Matrigel. Control segments exhibited robust sprouting and outgrowth, whereas stearic acid-treated rings showed no signs of endothelial proliferation, migration, or new vessel formation (Figure EV3E).

These findings demonstrate that stearic acid impairs critical steps in lymphangiogenesis, including endothelial migration and sprouting. While we focused on the tail model for in vivo relevance, future studies will be designed to explore systemic lymphatic beds and evaluate lymphatic repair beyond 28 days post-surgery to determine whether high SFAs exposure alters long-term recovery. This is now discussed in the revised manuscript.

Minor: "...insufficient to induce obesity" - how is this defined

Response: A significant difference in body weight between CD and HFD/HSFD groups was observed only at the final week of the 8-week dietary intervention. This duration was intentionally selected to avoid the development of overt obesity and its associated metabolic alterations, allowing us to isolate the effects of dietary fat composition. The ~16% increase in body weight observed in HFD- and HSFD-fed mice is considered modest and not indicative of obesity, particularly in female C57BL/6 mice, which are known to be relatively resistant to diet-induced obesity over moderate timeframes (J Diabetes Complications. 2021 Feb;35(2):107795).

Minor: in the abstract and intro it should state the source tissue for where a lower PUFA:SFA was identified. Likewise FABP4 in patients.

Response: We thank the reviewer for this suggestion. We have now clarified in both the Abstract and Introduction that the altered PUFA:SFA ratio and elevated FABP4 levels were identified in plasma samples.

Minor: saying that SFA causes these effects throughout is fine when specifically stating the outcome of the experiment, but because PA does cause the effects as well, in likely addition to other saturated fats not tested, it should not be made as a generalization in introducing or discussing the work.

Response: We have revised the manuscript to avoid overgeneralization of our findings.

10th Jul 2025

Dear Dr. Gibson,

Thank you for the submission of your revised manuscript to EMBO Molecular Medicine. I am pleased to inform you that we will be able to accept your manuscript pending the following final amendments:

1) Figures:

- Please remove Figure 8 as it is redundant with synopsis image. Remove also the callout in the text.
- We note that some images/panels are reused. e.g. Figure 6C is reused in Appendix Fig S11A and Figure EV4C is reused in Appendix Fig S18A. Please cite in the respective figure legend every reused image/panel.
- Please separate Parp1 and cPARP-1 Western blot images in Appendix Figure S13A.

2) Please address all comments suggested by our data editors listed below:

o Figure legends:

1. Please note that the figure 2 is mislabeled as figure 1 in the manuscript. Please check all figure numbers in the legends of the manuscript.
2. Please note that the exact p values are not provided in the legends of figures 1A-D; 3A-F; 4B-E; 5A, B, D-I; 6A, B, E, F; 7A, B, D-I; expanded view figures(s) 1 A-H; 2A, B, D-I; 3B, D; 4B, D, E, F; 5C, D; appendix expanded view figures(s) 1A, B; 2B, C; 4A, B; 5C, 6A-D; 7B, D, F; 9A, EV11, EV12 A-H; EV13 B, EV14 A, B; EV15 B, EV16 A-H.
3. Please note that information related to n is missing in the legends of figures 2E-J; appendix expanded view figure(s) 1A, B; 2A-D; EV3 B, 4A, B; 5A-D; 6A-D; 7B, D, F; 8A, EV11, EV12 A-H; EV13 B, EV14 A, B; EV15B, EV16 A-H.
4. Please note that the scale bar needs to be defined for figures EV5 A, B.
 - Please correct the heading from "Materials and Methods" to "Methods"
 - Please place the Data Availability section at the end of the Methods.
 - Please add callouts for the panels of Fig EV1 and EV5. A callout is left for a "Suppl. Table 2", please correct. Please correct the callouts for the Appendix Figures to "Appendix Figure S1" etc.
 - In Methods, provide the statement that informed consent was obtained from all human subjects and that the experiments conformed to the principles set out in the WMA Declaration of Helsinki and the Department of Health and Human Services Belmont Report.
 - In Methods, provide antibody dilutions that were used for each antibody.
 - Please remove the text about BioRender from the figure legend and add it in a dedicated "Graphics" section to the Methods, following this format:

Graphics:

(some of the... OR Figure #... OR synopsis) Graphics were created with BioRender.com.

- Indicate in legends exact n and exact p values, not a range, along with the statistical test used. To keep the figures "clear" some authors found providing an Appendix table Sx with all exact p-values preferable. You are welcome to do this if you want to.
- Author contributions: Please remove it from the manuscript and specify author contributions in our submission system. CRedit has replaced the traditional author contributions section because it offers a systematic machine-readable author contributions format that allows for more effective research assessment. You are encouraged to use the free text boxes beneath each contributing author's name to add specific details on the author's contribution. More information is available in our guide to authors:

<https://www.embopress.org/page/journal/17574684/authorguide#authorshippinguidelines>

- Please replace current text in data availability statement with the following: This study includes no data deposited in external repositories.

3) Appendix: Please correct the nomenclature to "Appendix Table S1" and "Appendix Table S2", and "Appendix Figure S1" etc. Place Appendix Table S2 after Appendix Table S1 and add page numbers to the table of contents.

4) Synopsis:

- Synopsis image: Please resize the image to 550 px-wide x 200-600 pixels high and upload it as a high-resolution jpeg file. Also, please adjust font size in the image, so that the text is readable in the new format.
- Please check your synopsis text and image before submission with your revised manuscript. Please be aware that in the proof stage minor corrections only are allowed (e.g., typos).

5) As part of the EMBO Publications transparent editorial process initiative (see our Editorial at

<http://embomolmed.embopress.org/content/2/9/329>), EMBO Molecular Medicine will publish online a Review Process File (RPF) to accompany accepted manuscripts. This file will be published in conjunction with your paper and will include the anonymous referee reports, your point-by-point response and all pertinent correspondence relating to the manuscript. Let us know whether you agree with the publication of the RPF and as here, if you want to remove or not any figures from it prior to publication. Please note that the Authors checklist will be published at the end of the RPF.

6) Please provide a point-by-point letter INCLUDING my comments as well as the reviewer's reports and your detailed responses (as Word file).

I look forward to reading a new revised version of your manuscript as soon as possible.

Yours sincerely,

Zeljko Durdevic

Zeljko Durdevic
Senior Editor
EMBO Molecular Medicine

*** Instructions to submit your revised manuscript ***

1) a .docx formatted version of the manuscript text (including Figure legends and tables)

2) Separate figure files*

3) supplemental information as Expanded View and/or Appendix. Please carefully check the authors guidelines for formatting Expanded view and Appendix figures and tables at <https://www.embopress.org/page/journal/17574684/authorguide#expandedview>

4) a letter INCLUDING the reviewer's reports and your detailed responses to their comments (as Word file).

5) The paper explained: EMBO Molecular Medicine articles are accompanied by a summary of the articles to emphasize the major findings in the paper and their medical implications for the non-specialist reader. Please provide a draft summary of your article highlighting

6) Author contributions: the contribution of every author must be detailed in a separate section.

7) EMBO Molecular Medicine now requires a complete author checklist

(<https://www.embopress.org/page/journal/17574684/authorguide>) to be submitted with all revised manuscripts. Please use the checklist as guideline for the sort of information we need WITHIN the manuscript. The checklist should only be filled with page numbers where the information can be found. This is particularly important for animal reporting, antibody dilutions (missing) and exact values and n that should be indicated instead of a range.

8) Every published paper now includes a 'Synopsis' to further enhance discoverability. Synopses are displayed on the journal webpage and are freely accessible to all readers. They include a short stand first (maximum of 300 characters, including space) as well as 2-5 one sentence bullet points that summarise the paper. Please write the bullet points to summarise the key NEW findings. They should be designed to be complementary to the abstract - i.e. not repeat the same text. We encourage inclusion of key acronyms and quantitative information (maximum of 30 words / bullet point). Please use the passive voice. Please attach

these in a separate file or send them by email, we will incorporate them accordingly.

You are also welcome to suggest a striking image or visual abstract to illustrate your article. If you do please provide a jpeg file 550 px-wide x 300-600px high.

9) A Conflict of Interest statement should be provided in the main text

10) Please note that we now mandate that all corresponding authors list an ORCID digital identifier. This takes <90 seconds to complete. We encourage all authors to supply an ORCID identifier, which will be linked to their name for unambiguous name identification.

Currently, our records indicate that the ORCID for your account is 0000-0003-0119-732X.

Link Not Available

11) Include a Reagents and Tools Table as part of the Methods section, which can be downloaded from our author guidelines (<https://www.embopress.org/page/journal/17574684/authorguide#structuredmethods>)

Photos 400-800 DPI

*Additional important information regarding figures and illustrations can be found at

<https://bit.ly/EMBOPressFigurePreparationGuideline>. See also figure legend preparation guidelines:

<https://www.embopress.org/page/journal/17574684/authorguide#figureformat>

***** Reviewer's comments *****

Referee #3 (Comments on Novelty/Model System for Author):

Lipid metabolism or targeting diet in lymphedema is not a novel idea, but the findings support patient efforts.

Referee #3 (Remarks for Author):

Combined with the responses to the other reviewers, my major concerns have been alleviated.

The authors addressed the remaining editorial issues.

21st Jul 2025

Dear Dr. Gibson,

We are pleased to inform you that your manuscript is accepted for publication and is now being sent to our publisher to be included in the next available issue of EMBO Molecular Medicine.

Zeljko Durdevic
Senior Editor
EMBO Molecular Medicine
